# A stabilized MERS-CoV spike ferritin nanoparticle vaccine elicits robust and protective neutralizing antibody responses

Abigail E. Powell [1] ✉, Hannah Caruso [1], Soyoon Park [1], Jui-Lin Chen[1], Jessica O'Rear[1], Brian J. Ferrer [1], Daniel J. Stieh [1], Adam M. Weiss [1], David M. Belnap [2], Audrey Walker[3], Anneliese Bruening[3], Airn Hartwig[3], Kaitlin R. Sprouse [4,5], Amin Addetia[4], Abeer N. Alshukairi[6,7], Vida Ahyong[1], Cristy S. Dougherty[1], David Veesler [4,5], Richard Bowen [3], Julie E. Ledgerwood[1], Michael S. Kay [1,8], Payton A.-B. Weidenbacher[1] & Brad A. Palanski[1]

Middle East respiratory syndrome coronavirus (MERS-CoV) was identified as a human pathogen in 2012 and causes ongoing sporadic infections and outbreak clusters. Despite case fatality rates (CFRs) of over 30% and considerable pandemic potential, a safe and efficacious vaccine has not been developed. Here we report the design, characterization, and preclinical evaluation of MERS-CoV antigens. Our lead candidate comprises a stabilized spike displayed on a self-assembling ferritin nanoparticle that can be produced from a high-expressing, stable cell pool. This vaccine elicits robust MERS-CoV pseudovirus and authentic virus neutralizing antibody titers in BALB/c mice. Immunization of male non-human primates (NHPs) with one dose of Alhydrogel-adjuvanted vaccine elicited a $> 10^3$ geometric mean titer of pseudovirus neutralizing antibodies that was boosted with a second dose. Sera from these NHPs exhibited cross-reactivity against spike-pseudotyped lentiviruses from MERS-CoV clades A, B, and C as well as a distant pangolin merbecovirus. In human DPP4 transgenic mice, immunization provided dose-dependent protection against MERS-CoV lethal challenge, and in an established alpaca challenge model using female alpacas, immunization fully protected against MERS-CoV infection. This MERS-CoV nanoparticle vaccine is a promising candidate for clinical advancement to protect at-risk individuals and for future use in a potential outbreak setting.

The emergence of three human pathogenic betacoronaviruses since 2002 has demonstrated the substantial, continuing threat that coronaviruses pose to the human population. MERS-CoV, the causative agent of Middle East respiratory syndrome, was first identified in 2012 and has since caused 2627 laboratory-confirmed cases and 947 deaths (36% CFR)[1]. MERS-CoV likely originated in bats and is transmitted from its intermediate host, dromedary camels, to humans[2–5]. In some instances, MERS-CoV undergoes human-to-human transmission, most

[1]Vaccine Company, Inc., South San Francisco, CA, USA. [2]School of Biological Sciences and Department of Biochemistry, University of Utah, Salt Lake City, UT, USA. [3]Department of Biomedical Sciences, Colorado State University, Fort Collins, CO, USA. [4]Department of Biochemistry, University of Washington, Seattle, WA, USA. [5]Howard Hughes Medical Institute, Seattle, WA, USA. [6]College of Medicine, Alfaisal University, Riyadh, Kingdom of Saudi Arabia. [7]Department of Medicine, King Faisal Specialist Hospital and Research Centre, Jeddah, Kingdom of Saudi Arabia. [8]Department of Biochemistry, University of Utah School of Medicine, Salt Lake City, UT, USA. ✉e-mail: apowell@vax.co

notably in healthcare settings, and continues to be an ongoing threat as highlighted by nine recent cases in 2025, two of which were fatal, in the Kingdom of Saudi Arabia[1,6]. There are currently no prophylactic vaccines or therapeutic interventions approved for use against MERS. The ongoing morbidity and mortality associated with MERS, the high prevalence of MERS-CoV in dromedary camel populations, and the potential for increased human-to-human transmission indicate a pressing need for a safe and effective MERS vaccine. The availability of such a vaccine would be useful as a routine prophylactic immunization for at-risk individuals such as healthcare workers, camel industry workers, and travelers to endemic regions. The vaccination of those at highest risk of MERS, due to occupation or travel, could mitigate the risk of a future pandemic and would provide proof-of-principle for deployment of this vaccine for broad use if it became needed.

MERS-CoV is a single-stranded RNA virus which displays a single surface protein, spike, responsible for host-cell recognition and viral entry[7–9]. MERS-CoV spike is a heavily glycosylated trimeric class I viral fusion protein that is expressed as a single polypeptide and cleaved via host cell proteases into $S_1$ and $S_2$ subunits, which remain non-covalently assembled following proteolysis[8]. The MERS-CoV spike receptor binding domain (RBD) in the $S_1$ subunit recognizes dipeptidyl peptidase 4 (DPP4)[10]. Following binding, the virus is endocytosed, at which time a secondary proteolytic cleavage event occurs at the $S_2'$ site within the $S_2$ subunit to allow viral membrane fusion to proceed[8]. Fusion can also occur at the viral membrane via TMPRSS2-mediated $S_2'$ cleavage if the virus has been precleaved at the $S_1/S_2$ site[11].

Antibodies targeting the spike protein following MERS-CoV infection demonstrate neutralizing activity[12–15]. Thus, the spike protein has been an important target for experimental MERS vaccine candidates[16–18]. Several vaccine platforms, including viral-based, subunit, and DNA have been evaluated in preclinical studies, some of which elicited MERS-CoV neutralizing antibodies and protected from viral challenge in preclinical models[19]. A few MERS-CoV vaccine candidates have been tested in early-stage clinical trials; however, no candidates have progressed to late-stage clinical development[20–24]. Therefore, advancing additional candidates towards clinical evaluation is critical.

Similar to other class I viral fusion proteins, the MERS-CoV spike trimer is a metastable protein poised to convert from the prefusion to the postfusion state upon receptor binding and cell entry[8]. Prior work has demonstrated that betacoronavirus spike proteins can be stabilized by mutating two key residues between the first heptad repeat (HR1) and second heptad repeat (HR2) to proline[25,26]. Installation of these two, as well as an additional four proline mutations in the SARS-CoV-2 spike, increased protein expression and led to enhanced immunogenicity, presumably as a result of improved stability of the prefusion conformation[26,27]. We therefore based our MERS-CoV antigen design on a stabilized spike containing a mutated S1/S2 cleavage site and 3 proline mutations (Fig. 1A). Structural studies of the MERS-CoV spike indicate that the region at the base of the trimer prior to the transmembrane domain (residues 1230-1294) is likely flexible, as it is unresolved via cryo-EM (Fig. 1B)[28]. We thus hypothesized, based in part on our prior betacoronavirus antigen design[29], that removal of this flexible region could facilitate stabilized presentation of the spike in a conformation that may be more conducive to eliciting neutralizing antibodies. Though this region of the $S_2$ domain can be the target of neutralizing antibodies[30], there is also evidence that it contains linear epitopes that elicit non-neutralizing antibodies in the context of SARS-CoV-2[31–33], and we reasoned that it could be in MERS-CoV spike as well, further supporting its removal.

Protein subunit vaccines have been shown in several contexts to have favorable stability profiles and to elicit robust, durable protection in humans[34–36]. In the absence of potent adjuvants, one limitation of protein subunit vaccines is weak immunogenicity compared to viral-based vaccines[35]. To overcome this limitation, proteins can be multimerized on nanoparticle platforms to confer benefits including facilitation of B cell cross-linking and enhanced neutralizing antibody responses[29,35,37–41]. However, recent efforts to develop a nanoparticle-based vaccine displaying the MERS-CoV RBD or spike elicited similar neutralizing antibody responses after two doses relative to the prefusion-stabilized spike[18], illustrating an ongoing need to identify optimal platforms and critical design attributes tailored for each specific antigen.

Ferritin is a ubiquitous, naturally occurring, self-assembling protein that forms 24-subunit particles[42]. It contains a 3-fold axis of symmetry and is therefore an ideal platform for display of trimeric viral antigens like the MERS-CoV spike protein[37]. We and others have shown that displaying the SARS-CoV-2 spike protein on the surface of *H. pylori* ferritin presents the antigen in a favorable conformation to elicit robust and cross-reactive neutralizing antibody responses[29,43–46]. Therefore, we applied these same antigen design principles to the MERS-CoV spike protein to develop a vaccine against MERS. Notably, ferritin nanoparticles have been demonstrated to be an effective and safe antigen multimerization platform in preclinical and clinical studies[43,47,48], but to our knowledge have not advanced past Ph2 human trials. This is likely due in part to challenges in scaling their manufacturing and purification. As a step in overcoming this hurdle, here we produce a high-expressing stable CHO cell pool, develop scalable purification methods, and perform forced degradation and accelerated stability studies on our lead vaccine candidate. These data provide unique insight informing the identification of critical quality attributes and methods necessary for scalability and process development for highly complex protein vaccine antigens.

The results presented here comprise the design, biochemical characterization, stability profile, and immunogenicity assessment of MERS-CoV spike vaccine candidates. We demonstrate that our lead candidate, MERS-1227 ferritin nanoparticle (FNP), can be purified to homogeneity and retains antigenicity and nanoparticle structural integrity following heat stress at 37 °C for 14 days. We show that MERS-1227 FNP is robustly immunogenic in BALB/c mice, NHPs, and alpacas as determined by MERS-CoV authentic virus and pseudotyped lentiviral neutralization assays. In transgenic BALB/c mice expressing human DPP4, we observed dose-dependent protection from MERS-CoV lethal challenge following immunization with our lead MERS-CoV vaccine candidate. In an alpaca challenge model, we observed no detectable viral infection in alpacas immunized with Alhydrogel-adjuvanted MERS-CoV nanoparticles whereas 80% of the placebo-treated alpacas shed infectious virus following challenge. The robust stability, immunogenicity, and efficacy profile of this protein-based nanoparticle vaccine demonstrate optimal attributes for clinical development as a safe and efficacious vaccine against MERS.

## Results

### Design of two ferritin-based MERS-CoV spike vaccine candidates with varied ectodomain lengths

The MERS-CoV spike is a 1353 amino acid polypeptide that is cleaved into $S_1$ and $S_2$ subunits via proteolysis at residue 751 (Fig. 1A)[8]. The ectodomain of MERS-CoV spike extends to residue 1294. We hypothesized that truncation of the ectodomain would facilitate more optimal presentation of the spike when displayed on the surface of a ferritin nanoparticle, potentially improving stability and immunogenicity. To determine the optimal truncation position for the spike ectodomain, we designed two constructs based on the spike sequence from MERS-CoV clade B England 1 strain[28]. These constructs extended through either residue 1227 (MERS-1227) or 1236 (MERS-1236) to optimally align the distance between the base of the protomers to the N-terminal residue of the ferritin 3-fold axis (Fig. 1C). Following the ectodomain in both constructs, we included an SGG linker and residues 5-167 from *H. pylori* ferritin (Fig. 1A) with an N19Q mutation in the ferritin sequence to eliminate a potential N-linked glycosylation site[37].

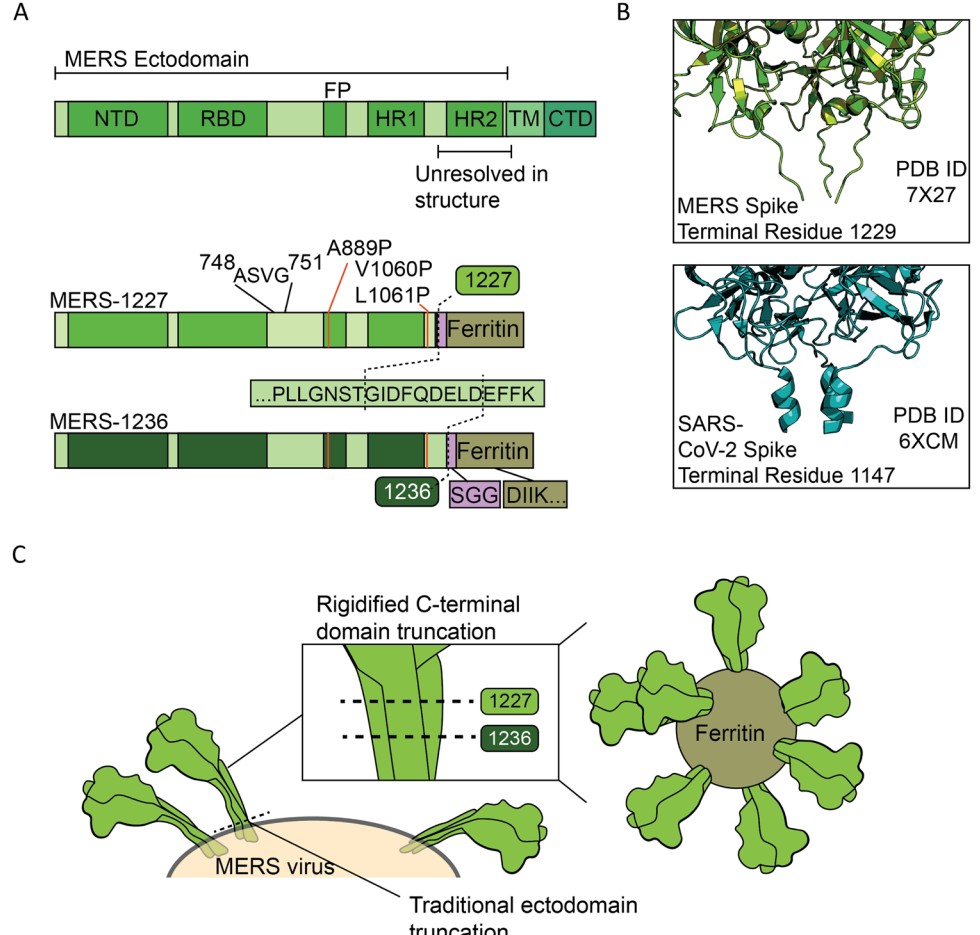

**Fig. 1 | Structure-informed design of two MERS-CoV spike ferritin nanoparticle vaccine candidates. A** Domain map of the MERS-CoV spike, containing the N-terminal domain (NTD), receptor-binding domain (RBD), fusion peptide (FP), heptad repeat 1 and 2 (HR1 and HR2), transmembrane domain (TM), and C-terminal domain (CTD). MERS-1227 and MERS-1236 contain a mutated furin cleavage site (ASVG) from 748-751, three stabilizing proline mutations (A889P, V1060P, L1061P), and ectodomain truncations after residue 1227 or 1236. MERS-1227 and MERS-1236 encode a flexible linker (SGG) followed by *H. pylori* ferritin starting at residue 5. **B** Structural comparison of the MERS-CoV spike (top, PDB 7X27) with the SARS-CoV-2 spike (bottom, PDB 6XCM) reveals a flexible domain in MERS-CoV whereas a short helix is present in SARS-CoV-2. **C** Visual representation of the impact of the MERS-1227 and MERS-1236 ectodomain truncations on the presentation of spike on the surface of the ferritin nanoparticle.

Additionally, we introduced three proline mutations guided by known SARS-CoV-2 2 P and 6 P variations that stabilize the spike protein[25–27]. We also mutated the $S_1/S_2$ furin cleavage site from RSVR to ASVG to prevent spike cleavage, as described in prior stabilized MERS-CoV constructs[25].

## MERS-1227 and MERS-1236 can be purified to homogeneity from high-expressing CHO cell pools and form stable nanoparticles with expected antigenicity

To express and characterize large quantities of MERS-1227 and MERS-1236 FNPs, we generated stable, high-expressing cell pools at ATUM utilizing their proprietary Leap-In™ transposase technology and CHOK1-derived mi:CHO-GS™ host cells[49]. The cell line development process yielded two stable cell pools with protein titers of approximately 2.7 (MERS-1227) and 3.2 (MERS-1236) grams per liter of cell culture in fed-batch supernatant harvested at day 14 as determined by SDS-PAGE quantitation (Fig. S1A). We purified the nanoparticles at a research-level scale to a high degree of homogeneity using anion exchange chromatography in flow-through mode, dialysis, and size-exclusion chromatography as reported previously for a related ferritin nanoparticle vaccine candidate[45]. Given the simplicity of this process, we utilized materials purified in

this manner for all subsequent analyses and immunizations described here. Additionally, we optimized a manufacturing process-applicable method for purification, which involved two bind and elute purification steps using an anion exchange membrane and a hydrophobic interaction chromatography (HIC) resin (Fig. S1B). Both steps result in enrichment of the MERS-CoV FNP, demonstrating the feasibility of future at-scale process development of these complex macromolecules.

We characterized the purity of MERS-1227 and MERS-1236 FNPs from the research-scale purification using SDS-PAGE analysis (Fig. 2A) and observed a pure species for both particles that migrates at the expected molecular weight of a MERS-CoV spike ferritin monomer (~150 kDa). Both proteins exhibited a slight decrease in migration when analyzed under reducing conditions, likely due to disruption of intra-protomer disulfide bonds. We further analyzed the purity and the size of the purified FNPs using size-exclusion chromatography multi-angle light scattering (SEC-MALS). This analysis revealed highly homogenous particles, without notable high or low molecular weight species, and MALS-calculated molecular weights of 4.27 ± 0.05 MDa (MERS-1227) and 4.23 ± 0.09 MDa (MERS-1236) (Fig. 2B). These observed molecular weights are slightly larger than those calculated from the MERS-1227 and MERS-1236 primary amino acid sequences

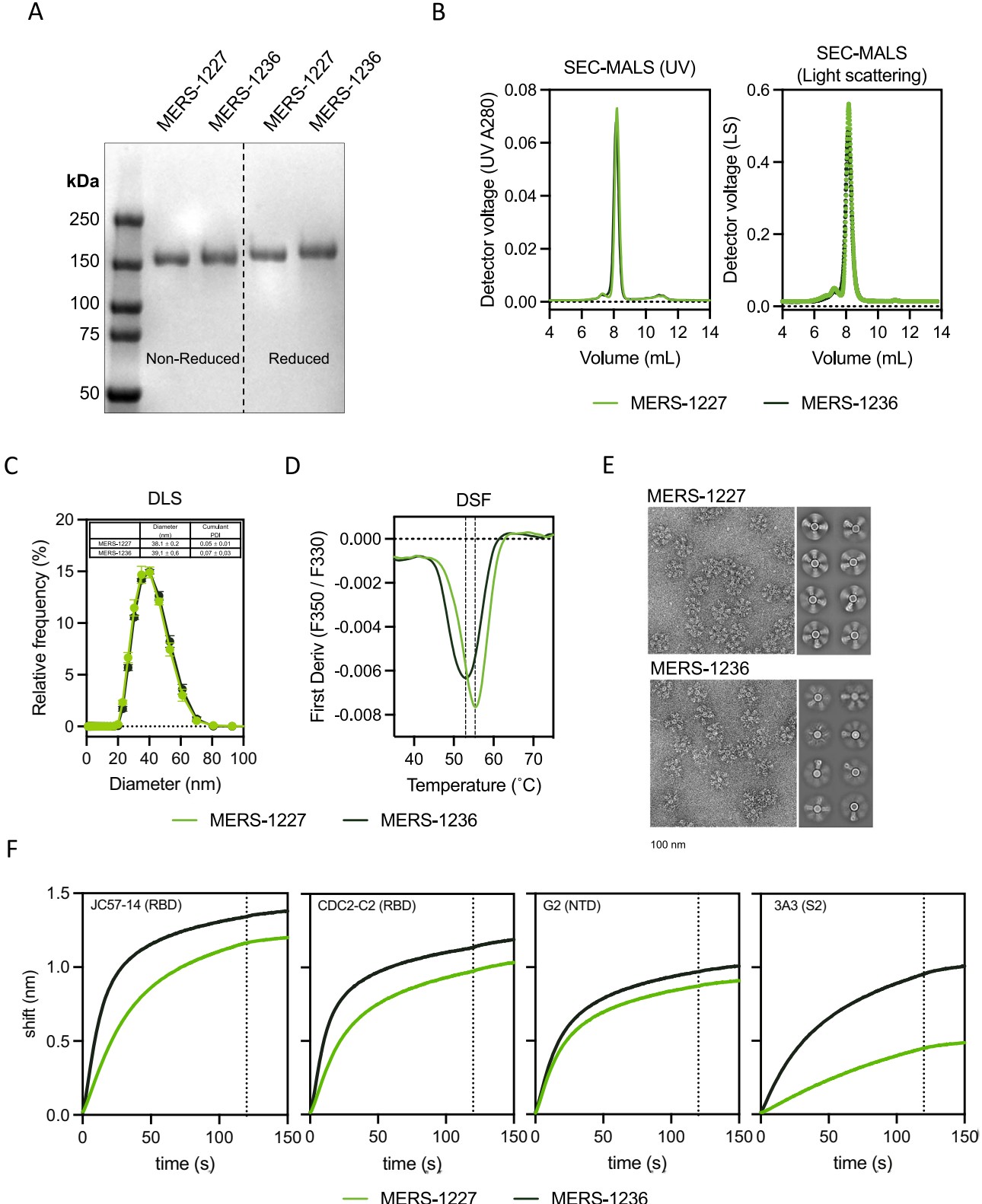

(3.6 and 3.7 MDa, respectively). We hypothesized that this difference was due to glycosylation of the MERS-CoV spike, which has previously been characterized[15,50–52]. To evaluate glycosylation of our MERS-CoV FNP proteins, we performed mass spectrometry-based peptide mapping to identify specific glycan modifications and to estimate their relative abundances at each glycosite (Supplementary Data 1). We observed extensive N-glycosylation and limited but detectable O-glycosylation throughout the MERS-1227 and MERS-1236 spikes but

not within the ferritin nanoparticle core (Fig. S2), consistent with prior analyses of purified MERS-CoV spike proteins[50–52].

Additional size characterization via dynamic light scattering (DLS) showed a diameter of 38.1 nm for MERS-1227 and 39.1 nm for MERS-1236 (Fig. 2C), consistent with the expected 10 nm diameter of an *H. pylori* ferritin nanoparticle with two ~14 nm MERS-CoV spikes projecting radially[8]. Interestingly, differential scanning fluorimetry (DSF) revealed distinct melting temperatures of 55.4 °C for MERS-1227 and

**Fig. 2 | MERS-1227 and MERS-1236 nanoparticles are comparable in size and homogeneity but have distinct stability, structural, and antigenicity features.** **A** SDS-PAGE analysis of MERS-1227 and MERS-1236 purified from 14-day fed-batch supernatants from stable CHO cell pools. SDS-PAGE analysis was run in technical duplicate with comparable results and one representative analysis is shown. **B** SEC-MALS using a Sepax SRT SEC-1000 column with light scattering measured by a Wyatt DAWN reveals that purified MERS-1227 and MERS-1236 are homogenous nanoparticles, with minimal aggregate species observed to the left of the nano-particle peak and no lower molecular weight species observed to the right. **C** DLS analysis of MERS-1227 and MERS-1236 reveals a particle diameter of ~40 nm. **D** DSF melting curves show that MERS-1236 has a melting temperature ~4 °C lower than MERS-1227 and a broader melting transition. **E** TEM micrographs (left) and 2D class averages (right) of MERS-1227 and MERS-1236 indicate differential spike presentation on the surface of the nanoparticles. Scale bar below micrographs represents 100 nm. TEM imaging and class averaging was conducted on two different batches of protein with comparable results. **F** Binding of MERS-1227 and MERS-1236 to MERS-CoV spike-specific mAbs measured by BLI. Binding assays were conducted in duplicate and the mean value is shown. Source data are provided in the Source Data file.

52.9 °C for MERS-1236, with MERS-1236 also exhibiting a broader melting profile (Fig. 2D). This result suggests that the 9 additional amino acids at the C-terminus of the ectodomain of MERS-1236 influence protein stability. However, the most striking difference between MERS-1227 and MERS-1236 was observed by negative stain transmission electron microscopy (TEM) (Fig. 2E), which showed that MERS-1236 spikes appear to exhibit modulated spike occupancy compared to MERS-1227. Collectively, these data indicate that MERS-1227 is a more stable functionalized nanoparticle, projecting the MERS-CoV spike protein radially from the ferritin core. Based on comparison of MERS-1227 and MERS-1236 via TEM and DSF, we elected to prioritize preclinical characterization of MERS-1227 and included MERS-1236 as a comparator when necessary. The structural and stability advantages of MERS-1227 suggested that it may exhibit fewer liabilities as a candidate for vaccine development.

To evaluate the antigenicity of MERS-1227 and MERS-1236, we utilized a panel of previously characterized monoclonal antibodies (mAbs) that bind to distinct epitopes across the MERS-CoV spike and used biolayer interferometry (BLI) to measure their binding to the nanoparticles. Within this panel, we included 11 S1-targeting antibodies[13,53–56], 8 known to bind the RBD and 3 known to bind out-side of the RBD, and 1 S2-targeting antibody[57] (Fig. 2F and S3, sequences provided in Supplementary Data 2). We observed that the antibodies targeting epitopes in the S1 domain outside of the RBD bind MERS-1227 and MERS-1236 equivalently, whereas RBD-directed anti-bodies exhibit enhanced binding to MERS-1236. Given that antibodies binding to S1 epitopes outside of the RBD (G2, FIB-H1, and CDC2-A2) bind to MERS-1227 and MERS-1236 equivalently, this suggests that non-RBD epitopes, such as those on the NTD, are presented on both pro-teins in the same manner. Regarding RBD-targeting epitopes, we hypothesize that the increased binding of RBD antibodies to MERS-1236 is related to the higher degree of flexibility of the spike on the surface of the nanoparticle, as evidenced by our TEM results. This flexibility may impact the degree to which RBDs are displayed in the "up" or "down" conformation on MERS-1236 compared to MERS-1227, which could impact binding of antibodies that target RBD epitopes, or the enhanced flexibility could increase accessibility to IgGs. Alter-natively, increased trimer flexibility in MERS-1236 could allow for antibody bridging between adjacent trimers on the same nanoparticle which may not occur on MERS-1227. Furthermore, the $S_2$-directed antibody, 3A3, binds at the trimer interface which is only accessible in an open conformation of SARS-CoV-2 spike that the trimer reversibly samples[58]. The increased binding of MERS-1236 to 3A3 is therefore consistent with a more flexible trimer and may suggest MERS-1236 samples an "open trimer" conformation more frequently. We con-firmed binding specificity by demonstrating that MERS-1227 and MERS-1236 do not bind to a SARS-CoV-2 specific antibody, casirivimab.

**MERS-1227 and MERS-1236 are stable over a range of pH and temperature conditions**
Low-pH treatment is commonly employed for viral inactivation during downstream process development, and knowledge of pH tolerance is helpful in optimizing chromatography steps. Therefore, we sought to understand the effect of low pH on MERS-1227 and MERS-1236 nanoparticle stability and antigenicity. We formulated purified MERS-1227 and MERS-1236 in pH 7.5 formulation buffer and held them in the pH-adjusted solutions at pH 2, 3, 3.5, and 4 for 2 h. Then, the nano-particles were dialyzed back into pH 7.5 buffer and evaluated using analytical SEC, DLS, DSF, and BLI (Fig. 3A and S4A-B). These analyses revealed that pH 2 treatment causes irreversible damage to both MERS-1227 and MERS-1236 as observed by all biophysical measurements, suggesting dissociation of the nanoparticle and/or denaturation of the MERS-CoV spike. BLI revealed that binding of an RBD mAb (CDC-C2) and an NTD mAb (G2) is retained following pH 2 treatment, which may suggest that antigenicity of the isolated protomer is retained to some degree fol-lowing nanoparticle dissociation. Treatments at pH 3, 3.5, and 4 do not notably affect particle structure or antigenicity.

Temperature stability is important in vaccine development with the goal of optimizing a modality and formulation such that cold-chain requirements can be minimized. This is especially relevant to a MERS vaccine, as the endemic regions where it would be distributed reg-ularly experience high ambient temperatures. To understand the thermostability of MERS-1227 and MERS-1236, we formulated purified materials in formulation buffer and stored them at 4, 22, or 37 °C for 14 days. Following these temperature holds, we evaluated the quality and stability of the FNPs using analytical SEC, DLS, DSF, and BLI (Fig. 3B and S4C-D). We confirmed that DLS, DSF, and BLI binding to these mAbs are appropriate metrics of protein integrity, as incubation for 30 min at 90 °C notably perturbed particle size (DLS) and fully ablated the melting curve (DSF) and binding to both mAbs (BLI) for both FNPs (Fig. 3B). By SEC analysis, we observed that both MERS-1227 and MERS-1236 remained stably associated as nanoparticles even after 14 days at 37 °C, albeit with a minor fraction of particles forming an aggregate species as evidenced by a peak at ~7.0 min that elutes prior to the nanoparticle peak at ~8.0 min. DLS showed no major change in the size of particles following heat-treatment, and DSF revealed no change in the primary melting peak following the temperature hold conditions, providing evidence for the conformational stability of the particles. We also found that both MERS-1227 and MERS-1236 retain antigenicity against an RBD-specific (CDC-C2) and an NTD-specific (G2) monoclonal antibody after 14 days at either 22 or 37 °C (Fig. 3B).

**MERS-1227 and MERS-1236 are more immunogenic compared to trimer alone in BALB/c mice following one or two doses**
We sought to determine the effect of spike multimerization on elici-tation of neutralizing antibodies by immunizing BALB/c mice (n = 5 or 10) with MERS-1227, MERS-1236, or the MERS-CoV full-length spike (S) ectodomain trimer (residues 1-1294, containing the 3 proline mutations and mutated furin site used in MERS-1227 and MERS-1236) with a GCN4 trimerization domain[59] as a comparator. We immunized the antigens at a total protein dose of 2 or 10 μg with 150 μg Alhydrogel (Fig. 4A) and collected serum at day 21 (following a single dose) and day 42 (following two doses). As shown in Fig. 4B, after a single dose, mice immunized with either 2 or 10 μg of MERS-1227 or MERS-1236 FNPs elicited ~2- to 3-fold higher levels of both anti-spike and anti-RBD IgG antibodies as compared to mice immunized with MERS-CoV S trimer. Following a second dose, mice across the three antigen groups

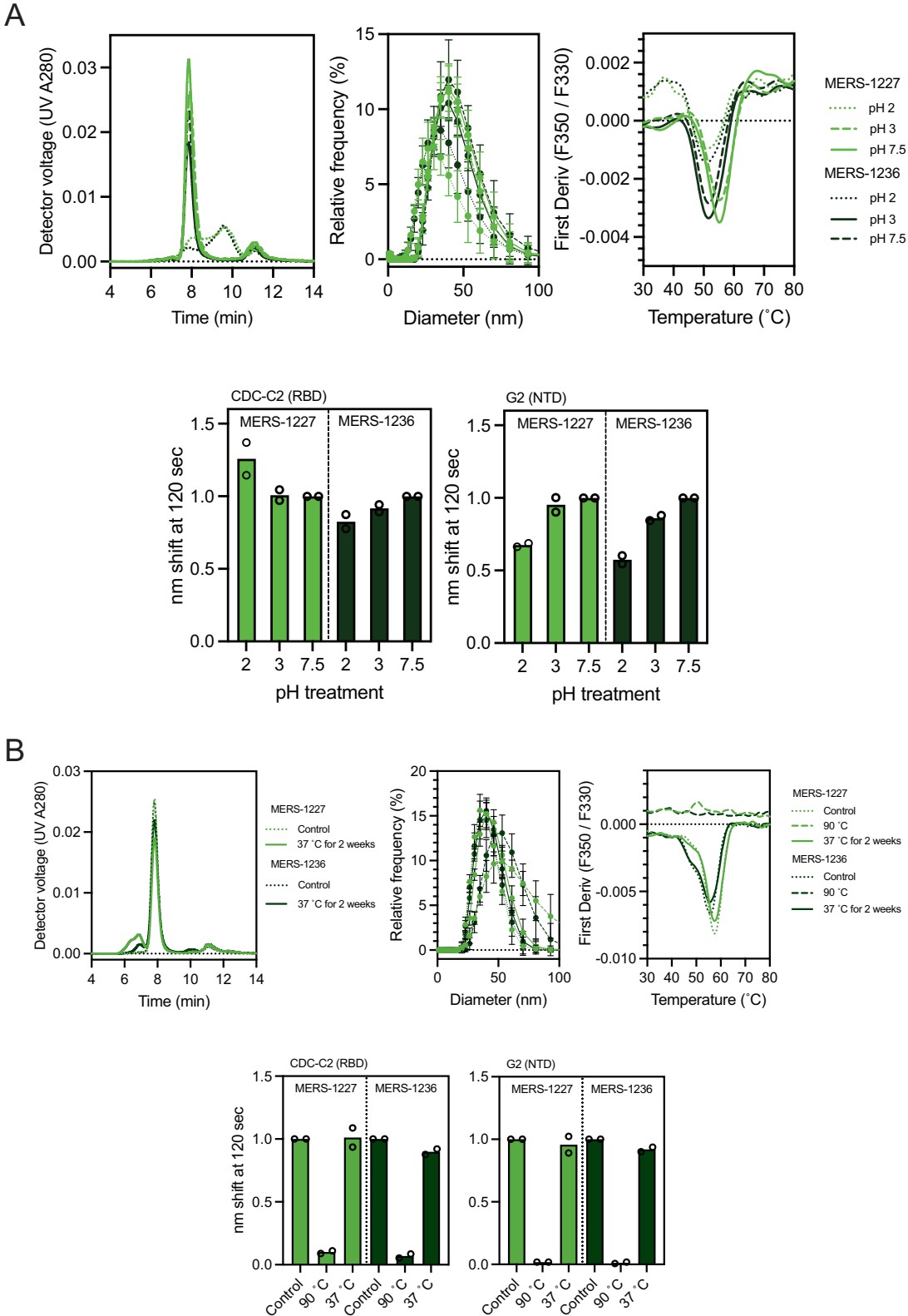

**Fig. 3 | pH and temperature stability of MERS-1227 and MERS-1236. A** SEC, DLS, DSF, and BLI results for MERS-1227 and MERS-1236 treated at pH 2, 3, or 7.5 for 2 h. **B** SEC, DLS, DSF, and BLI results for MERS-1227 and MERS-1236 treated at 37 °C for 2 weeks. A 30 min 90 °C treated control is shown for DLS, DSF, and BLI. For both (**A**) and (**B**) SEC, DLS, and DSF results were performed in experimental duplicate and one representative replicate is shown. DLS measurements are taken via 10 individual acquisitions from a single technical replicate; points represent the mean value and error bars represent the standard deviation of the individual acquisitions, with one representative replicate shown. BLI results were performed in experimental duplicate and normalized to the pH 7.5 condition. The height of the bar represents the mean of two experimental replicates and the circles represent the individual values from these replicates. Source data are provided in the Source Data file.

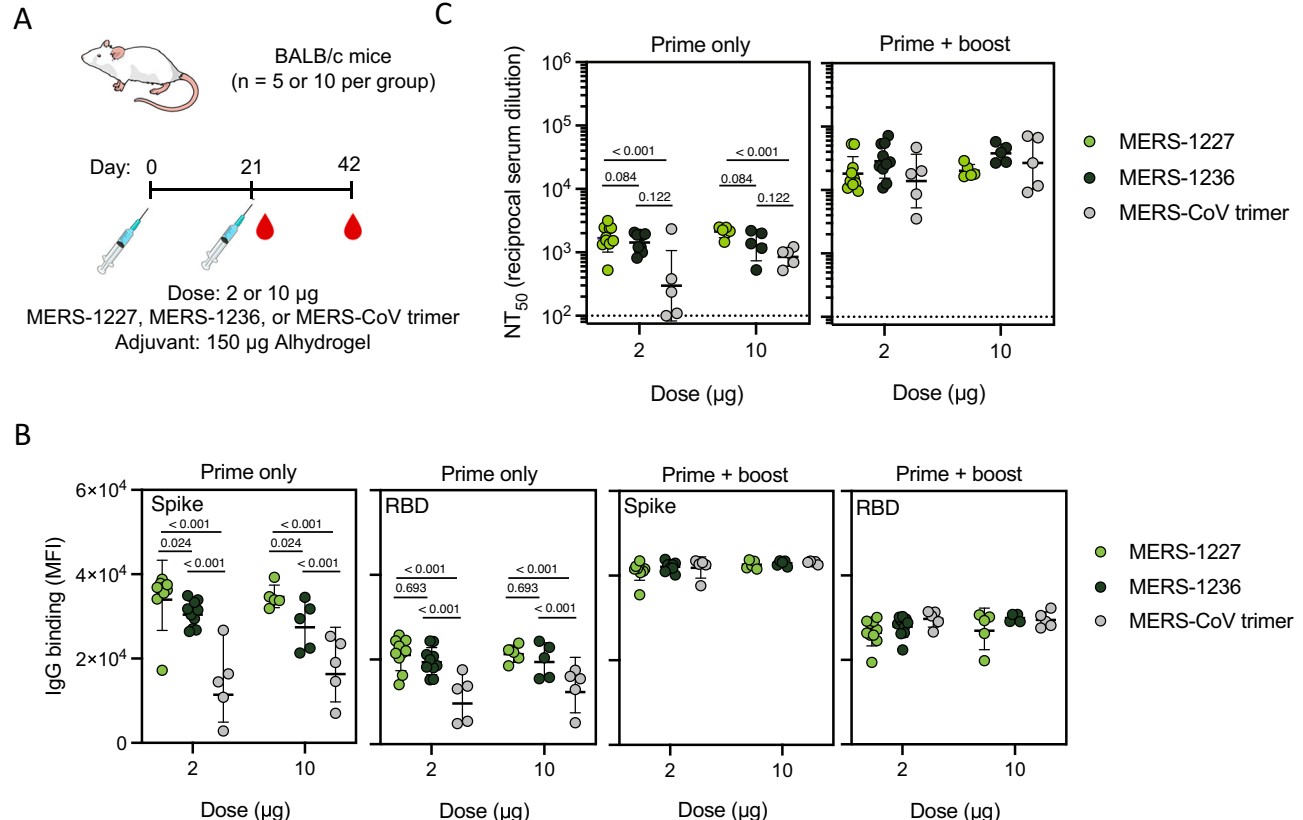

**Fig. 4 | Immunization of BALB/c mice with MERS-1227 or MERS-1236 results in more robust binding and neutralizing antibody responses compared to soluble trimer. A** Immunization schema (mouse image from bioart.niaid.nih.gov/bioart/279). MERS-1227 and MERS-1236 at the 2-μg dose level were immunized in groups of $n = 10$; all other antigens and doses were immunized in groups of $n = 5$. **B** Antigen-specific IgG binding responses were measured in technical duplicate at 1:100 dilution using a Luminex assay with beads coated with either soluble MERS-CoV trimer or MERS-CoV RBD. Each dot represents the average of the mean fluorescence intensity (MFI) value per mouse, lines represent geometric mean of the immunization group, and error bars represent the geometric SD. Statistical analysis was conducted using a two-way ANOVA with Tukey's multiple comparison test. No statistical differences within dose groups were observed at the prime + boost timepoint. **C** MERS-CoV spike-pseudotyped lentivirus neutralization GMTs observed following a prime only (day 21) and a prime + boost (day 42). Each point represents the $NT_{50}$ value for an individual mouse obtained by running pseudovirus neutralization analysis in quadruplicate (with the exception of one mouse in the 10 μg trimer group at day 21 in duplicate). Lines represent the GMT, and error bars represent the geometric SD. Statistical analysis was conducted using a two-way ANOVA with Tukey's multiple comparison test. No statistical differences within dose groups were observed at the prime + boost timepoint. Source data are provided in the Source Data file.

at all doses exhibited similar levels of MERS-CoV spike and RBD IgG (Fig. 4B, right panels).

We quantified neutralizing antibody titers using a luciferase-based assay[60] with lentivirus pseudotyped with MERS-CoV spike and HeLa cells stably expressing human DPP4 as the targets of infection. We validated our assay using MERS-CoV specific neutralizing antibodies with known $IC_{50}$ values[13,55]. The values obtained in our assay concorded well with those reported in the literature (Fig. S5A). As expected, no neutralization was detected with a negative control SARS-CoV-2 specific mAb, casirivimab. We also used convalescent serum from a camel challenged with MERS-CoV with a known $PRNT_{90}$ of 1:320 serum dilution as a benchmark for our pseudovirus neutralizing titers (Fig. S5B)[61]. In our pseudovirus assay, this camel serum sample inhibits 90% of viral entry ($IC_{90}$) at a serum dilution of 1:6810. This result indicates that our lentivirus-based pseudovirus assay exhibits higher sensitivity than a plaque-reduction neutralization assay using authentic MERS-CoV virus, as reported for other lentivirus-based neutralization assays compared to authentic viral comparators[62].

Following a single dose, we observed robust pseudovirus neutralizing titers of GMT ~ $10^3$ for MERS-1227 and MERS-1236 at all doses tested. Comparatively, MERS-CoV trimer elicited a lower and more heterogeneous antibody response at 2 μg and a more homogenous but still overall lower response at 10 μg. After a second dose, we

observed a boost in titers across all antigens and dose groups. Notably, the MERS-CoV S trimer groups show a higher degree of variability, especially at the 10-μg dose level, with a geometric standard deviation of 2.6 as compared to 1.3 and 1.4 for MERS-1227 and MERS-1236, respectively (Fig. 4C, right panel). We additionally tested pools of serum from the MERS-1227 and MERS-1236 groups and confirmed they neutralized authentic MERS-CoV in a plaque-reduction neutralization assay (Table S1). This initial mouse immunogenicity study revealed that the multimerized, stabilized MERS-CoV spike elicited a stronger neutralizing antibody response following one dose and a more homogeneous response following two doses compared to soluble trimer. Subsequently, we performed an additional dose de-escalation study with Alhydrogel-adjuvanted MERS-1227 and found that it elicited a neutralizing antibody response at 0.016 μg, the lowest dose tested, following a prime and a boost (Fig. S6).

## MERS-1227 and MERS-1236 elicit robust neutralizing antibody responses in NHPs following a single dose that increase and persist at least several months following a second dose

To evaluate MERS-1227 and MERS-1236 immunogenicity in an NHP model, we immunized cynomolgus macaques ($n = 5$ per antigen) with 50 μg FNP (total protein dose) adjuvanted with 750 μg Alhydrogel and

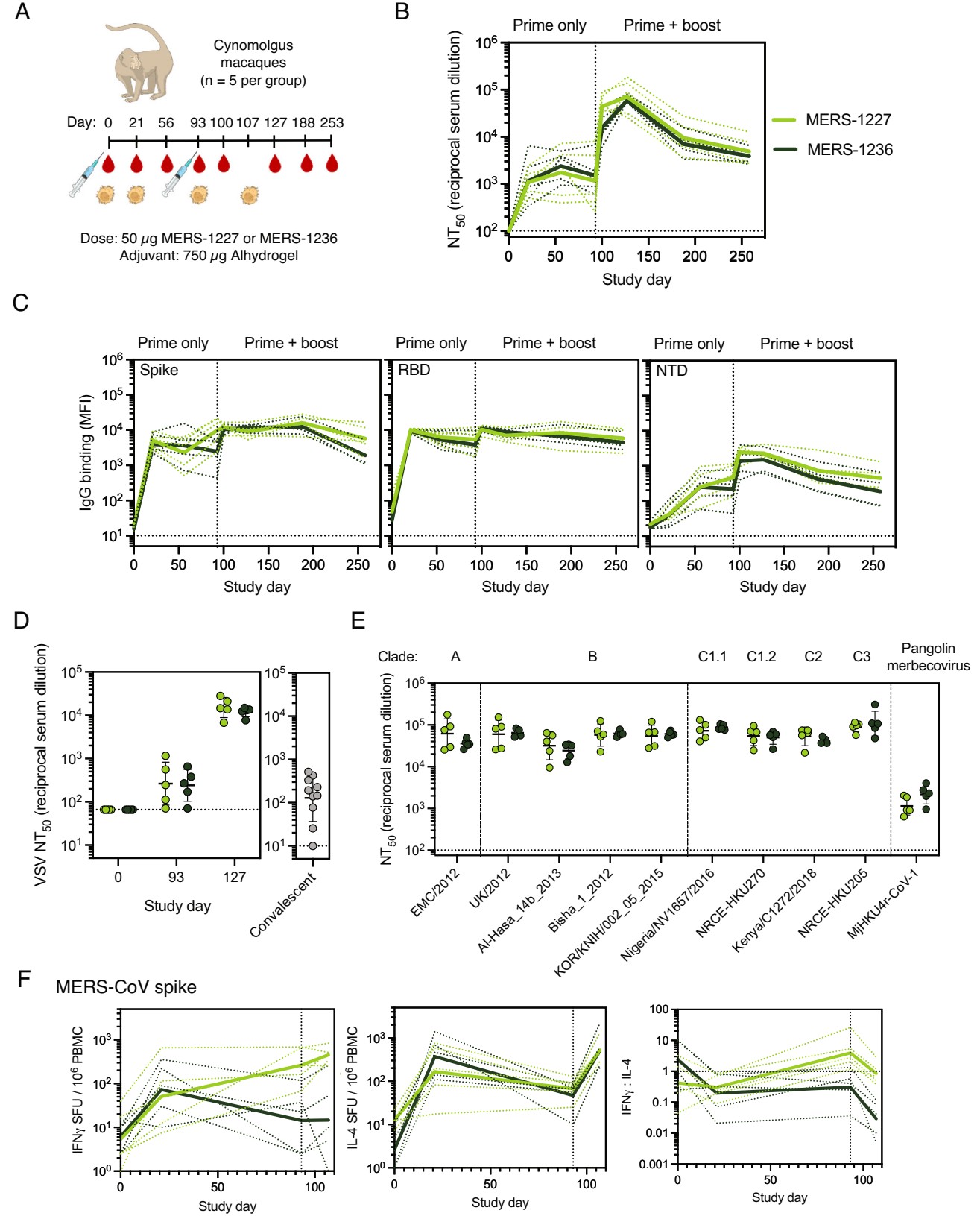

measured the magnitude, durability, and breadth of the antibody response following one and two doses (Fig. 5 and Table S2). Following a single immunization, we observed pseudovirus neutralizing antibody titers at or greater than reciprocal serum dilution ($NT_{50}$) of $10^3$ for all animals in both vaccine groups (Fig. 5B). Importantly, we did not observe notable waning of these titers over the three-month period

following the prime vaccination. We immunized the NHPs with a homologous boost at study day 93 and observed a robust increase of greater than 10-fold in pseudovirus neutralizing titers at study day 100 that continued to increase to study day 127. At study day 253 (160 days post-boost), titers remained higher than those observed following a single dose. Importantly, the rate of decay observed between days 180

**Fig. 5 | NHPs immunized with Alhydrogel-adjuvanted MERS-1227 and MERS-1236 elicit robust, broad, and durable neutralizing antibody responses. A** NHP immunization schedule with serum and PBMCs collected on days indicated. Each immunization group contained $n = 5$ NHPs. NHP image from bioart.niaid.nih.gov/bioart/388. **B** MERS-CoV spike pseudotyped lentivirus neutralization GMTs following a prime and a boost. The solid lines represent the average GMTs of each group and the dashed lines represent the GMTs by individual NHP; individual GMT values were obtained by running serum from each animal in quadruplicate. **C** Antigen-specific IgG binding responses were measured in duplicate at 1:100 dilution using a Luminex assay with beads coated with either soluble MERS-CoV trimer, MERS-CoV RBD, or MERS-CoV NTD. The solid line represents the average MFI per group, and the dashed lines represent the MFIs per individual NHP. **D** Neutralizing titers from NHP serum collected at study days 0, 93, and 127 assessed in a VSV spike-pseudotyped neutralization assay. Titers are compared to those obtained from MERS convalescent patient samples (gray circles). Dashed lines indicate LOQ (1:66 dilution for NHP assessment and 1:10 for convalescent serum assessment). Individual points represent the $NT_{50}$ value from one NHP or one patient obtained from running each sample in biological replicate with two different batches of pseudovirus. Lines represent GMT and error bars represent geometric SD. **E** NHP serum collected ~5 weeks following a second immunization (study day 127) neutralizes a panel of lentiviruses pseudotyped with spike sequences from MERS-CoV clades **A**–**C**, and a pangolin merbecovirus (MjHKU4r-CoV-1). Each circle represents the $NT_{50}$ value for an individual NHP obtained by running pseudovirus neutralization analysis in quadruplicate. Lines represent the GMT of the immunization group and error bars represent the geometric SD. (**F**) IFNγ (left) and IL-4 (right) responses to MERS-CoV spike trimer were quantified via ELISpot. The solid lines represent the average values of each group and the dashed lines represent the GMTs by individual NHP. The reported values for each individual NHP were obtained using the median of two technical replicates. Source data are provided in the Source Data file.

and 253 was decreased compared to that observed between days 127 and 180, which is suggestive of the potential for a durable response as has been reported for other alum-adjuvanted protein-based vaccines[63]. To further characterize the response, we used a Luminex panel containing MERS-CoV spike, RBD, and NTD, and evaluated total IgG levels against these antigens. We observed similar spike and RBD IgG levels, with slightly more waning of the spike titers observed between study days 188 and 253. Interestingly, spike and RBD binding titers decrease less quickly than pseudovirus neutralizing titers (Fig. 5, compare B to C).

To understand how antibody responses in NHPs compare to those elicited in the context of human infection with MERS-CoV, we used VSV pseudotyped with MERS-CoV EMC/2012 S to evaluate neutralizing antibody titers from NHPs at study days 93 and 127 side-by-side with a panel of MERS patient serum samples. As shown in Fig. 5D (full neutralization curves shown in Fig. S7A), following a single dose (at 93 days), GMTs were approximately 2-fold higher than the convalescent samples ($NT_{50}$ 255 compared to 130 serum dilution) which improve to over 100-fold higher after two doses ($NT_{50}$ 14,058 compared to 130 serum dilution).

To characterize the breadth of the antibody response to known MERS-CoV strains, we established a panel of MERS-CoV spike-pseudotyped lentiviruses comprising clades A, B (currently circulating in humans and the vaccine strain in this study), C, and a distantly related pangolin merbecovirus, MjHKU4r-CoV-1[64,65]. We observed robust neutralization of all MERS-CoV clades in the panel (Fig. 5E), including clade C strains which are known to circulate in camels but not humans[64]. Remarkably, we also observed neutralization of MjHKU4r-CoV-1, which only shares a 66% amino acid sequence identity with the spike of England 1 MERS-CoV, albeit at reduced potency relative to other strains tested[65]. Importantly, we did not observe neutralization of the MjHKU4r-CoV-1 pseudotyped lentivirus with pre-immune NHP serum (Fig. S7B). Pseudovirus neutralization, breadth, and total IgG levels were similar between MERS-1227 and MERS-1236 immunized NHPs. Overall, the results of this study demonstrate that adjuvanted MERS-1227 and MERS-1236 elicit a robust, durable, and broad response in NHPs that can be boosted following a second dose administered at a 3-month interval, which is consistent with other effective vaccine regimens[66].

To understand cellular immune responses to MERS-1227 and MERS-1236, we collected peripheral blood mononuclear cells (PBMCs) from vaccinated NHPs at study days 0, 21, 93, and 107 (Fig. 5A). We conducted ELISpot analysis to assess Th1 and Th2 (interferon gamma (IFNγ) and IL-4, respectively) responses to both MERS-CoV spike trimer and *H. pylori* ferritin (Fig. 5F and S7C). IFNγ and IL-4 responses to both spike and ferritin are primed and can be boosted following one and two doses of vaccine. Interestingly, spike and ferritin responses are similar between NHPs immunized with MERS-1227 and MERS-1236 following a single dose, but NHPs vaccinated with MERS-1227 exhibit enhanced IFNγ to both spike and ferritin following two doses. IL-4 responses are similar across the two vaccination groups at all timepoints tested. After boosting, the ratio of the IFNγ to the IL-4 response for both MERS-CoV spike and ferritin is below 1 for MERS-1236 vaccinated animals implying a Th2-biased response, and near 1 in MERS-1227 vaccinated animals, suggesting balanced Th1/Th2 immunity. Of note, preclinical studies with SARS1 vaccine candidates have previously demonstrated that immunization inducing a Th2 dominated response resulted in lung immunopathology following viral challenge[67]. Therefore, while the response observed in NHPs was robust, the potential for a Th2-biased response to MERS-1236 indicates that evaluation of other adjuvants may be key to balancing high neutralizing antibody responses while reducing potential for Th2-induced immunopathology.

## Vaccination with MERS-1227 or MERS-1236 protects from viral shedding in alpacas

Alpacas have been established as a surrogate infection model for camels, the predominant intermediate animal host of MERS-CoV. Unlike camels, alpacas do not develop rhinorrhoea (nasal discharge) upon MERS-CoV infection, but they do shed infectious virus that is detectable in nasal turbinates[61,68]. We immunized alpacas ($n = 5$ per group) at two dose levels (20 or 200 μg) of MERS-1227 or MERS-1236 adjuvanted with 500 μg Alhydrogel using a prime-boost regimen (Fig. 6A and Table S2). As a placebo control, an additional group of alpacas ($n = 5$) were dosed with 500 μg Alhydrogel at the same time points. Three weeks following the boost, (study day 41), all vaccinated alpacas had detectable pseudovirus neutralizing titers in their sera (Fig. 6B). Furthermore, all alpacas vaccinated at both dose levels of MERS-1227 and the 200-μg dose of MERS-1236 had detectable neutralizing titers in an authentic virus assay (Fig. 6C).

On study day 42, alpacas were challenged by intranasal inoculation with live MERS-CoV virus (clade A, EMC/2012) in a BSL3 containment facility, as previously reported[68]. On days 1-5 and 7 post-challenge, nasal swabs were collected from each nare, and authentic virus neutralization titers were determined using a plaque assay[68]. Detectable levels of MERS-CoV were observed via nasal swab for at least one time point following challenge in 4/5 alpacas that received the Alhydrogel placebo, while authentic virus was not detected in any (0/20) of the alpacas that received either dose level of Alhydrogel-adjuvanted MERS-1227 or MERS-1236 (Fig. 6D). Taken together, these results demonstrate that immunization with Alhydrogel-adjuvanted MERS-1227 and MERS-1236 elicited protection from infection with authentic MERS-CoV.

At this stage of the preclinical development process, we elected to move forward with further characterization of one antigen. While the immunogenicity results in BALB/c mice, NHPs, and alpacas appeared similar between MERS-1227 and MERS-1236, both the decreased melting temperature and increased structural flexibility of MERS-1236 suggested to us that these could be important liabilities during vaccine manufacturing or long-term stability studies. For these reasons, we conducted further evaluations with MERS-1227.

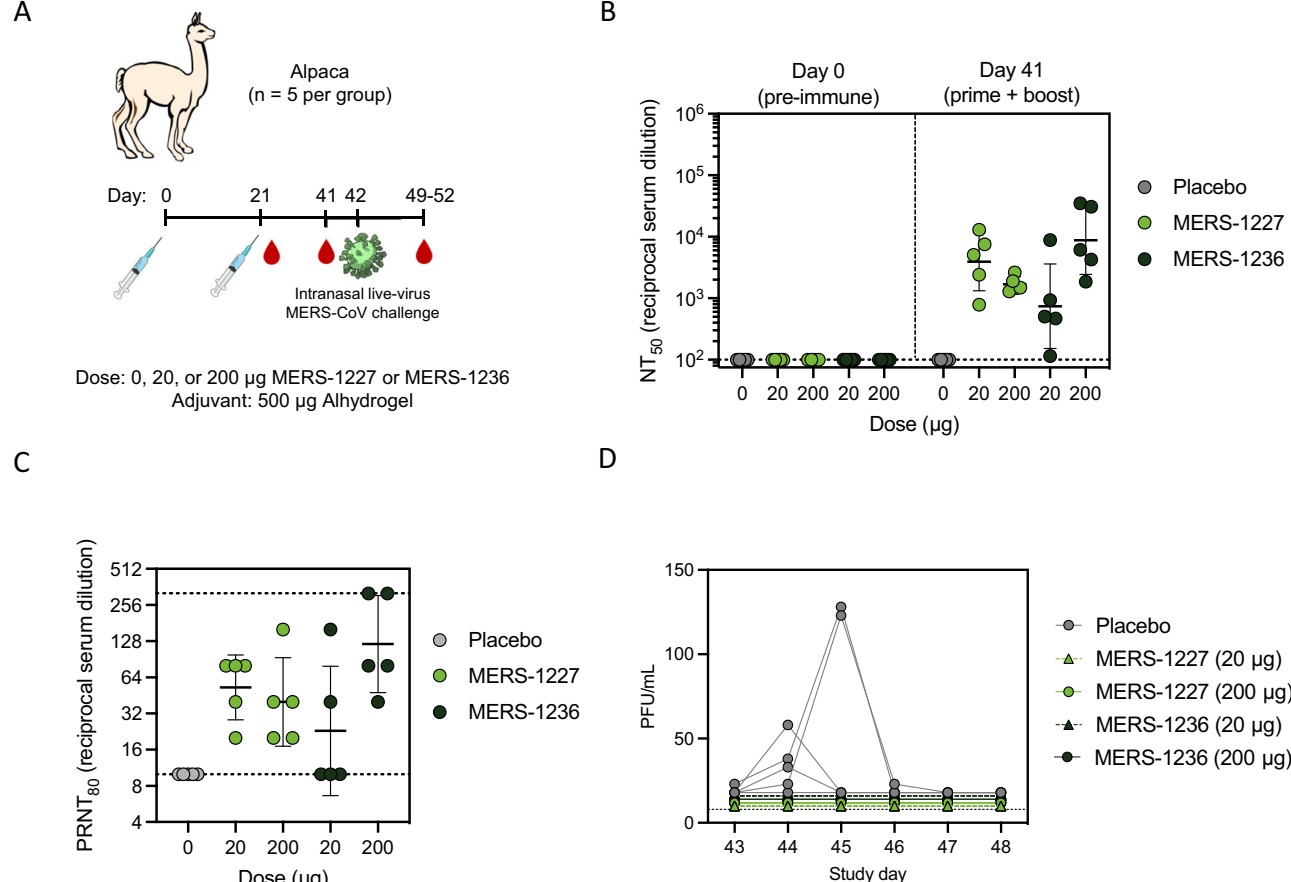

**Fig. 6 | Alpacas immunized with Alhydrogel-adjuvanted MERS-1227 and MERS-1236 are fully protected from MERS-CoV infection. A** Alpaca challenge study design and timeline. Each immunization group contained $n = 5$ alpacas. Alpacas were challenged at day 42 via intranasal inoculation with EMC/2012 MERS-CoV. **B** Pseudovirus neutralizing titers from alpacas at day 0 (pre-immune) and 41 following two immunizations. Each circle represents the $NT_{50}$ value for an individual alpaca obtained by running pseudovirus neutralization analysis in quadruplicate. Lines represent the GMT of the immunization group and error bars represent the geometric SD. **C** Live-virus neutralization obtained from a plaque reduction neutralization assay using day 41 serum samples following two immunizations. Each circle represents the $PRNT_{80}$ value for an individual alpaca obtained from a single assay run utilizing seropositive and seronegative camel sera as assay controls. Lines represent GMT and error bars represent geometric SD. **D** Quantitation of live MERS-CoV virus in nasal swabs obtained from alpacas from days 1–5 and 7 post viral challenge via plaque assay. Each circle represents the mean number of plaque forming units observed in an individual alpaca from independent swabs of the left and right nare per mL inoculum. Source data are provided in the Source Data file.

## The immune response to MERS-1227 can be tuned using different adjuvant combinations

While alum is the most widely used vaccine adjuvant with a long clinical history and can be readily procured for clinical development programs, alternative adjuvants have potential to enhance the immunogenicity and durability of protein subunit vaccines[69]. We therefore formulated MERS-1227 with a panel of adjuvants known to engage distinct immune pathways[69], including CpG (ODN2395), Quil-A + monophosphoryl lipid A (MPLA), AddaVax, and 3M-052 and probed the resulting immunogenicity in mice. Based on prior preclinical and clinical use of these adjuvants, we evaluated some co-formulated with Alhydrogel (Quil-A + MPLA, CpG, and 3M-052) as well as some without Alhydrogel (AddaVax, Quil-A + MPLA, and 3M-052). We formulated MERS-1227 with the various adjuvants and immunized BALB/c mice with a 0.4 µg dose of FNP protein followed by a homologous boost at day 21 (Fig. 7A).

An adjuvant was required to elicit a robust response, as unadjuvanted MERS-1227 hardly elicited detectable pseudovirus neutralization titers compared to all three doses of Alhydrogel tested (Fig. 7A). Titration of Alhydrogel in the absence of other adjuvants indicated that in mice, the dose response is maximal between 15 and 50 µg. Immunization with MERS-1227 formulated with AddaVax (similar to MF59[70] in clinically approved vaccines) did not improve titers compared to the mid- and high-dose Alhydrogel groups

following a single dose but did show a ~ 3-4 fold enhancement in titers following the boost.

The most striking improvement in immunogenicity was observed with Quil-A + MPLA (in the absence of Alhydrogel), which showed a 4-fold enhancement in titers over the 150 µg Alhydrogel group after one dose and a 22-fold enhancement after the second dose. While addition of Quil-A + MPLA to Alhydrogel improved responses compared to the matched 15 µg and 150 µg Alhydrogel-only groups, the strongest response to Quil-A + MPLA was observed in the absence of Alhydrogel. Addition of CpG (ODN2395) only conferred an immunogenicity improvement compared to Alhydrogel alone at the 15 µg Alhydrogel dose, with minimal difference between the 150 µg Alhydrogel +/- CpG groups at both timepoints. Interestingly, addition of 3M-052, a lipidated TLR7/8 agonist[71], did not result in increased neutralizing titers compared to the Alhydrogel only dose groups after a single dose but did show an improved response following a boost, most notably in the presence of either 15 or 150 µg Alhydrogel.

To evaluate the influence of adjuvants on the ratio of Th1/Th2 responses following immunization with MERS-1227, we bound biotinylated MERS-CoV spike trimer to streptavidin Luminex beads and quantified the levels of mouse IgG2a and IgG1 (Fig. S8). In mice, induction of IgG1 is associated with a Th2-biased response and IgG2a is associated with a Th1-biased response[72]. Mice immunized with MERS-

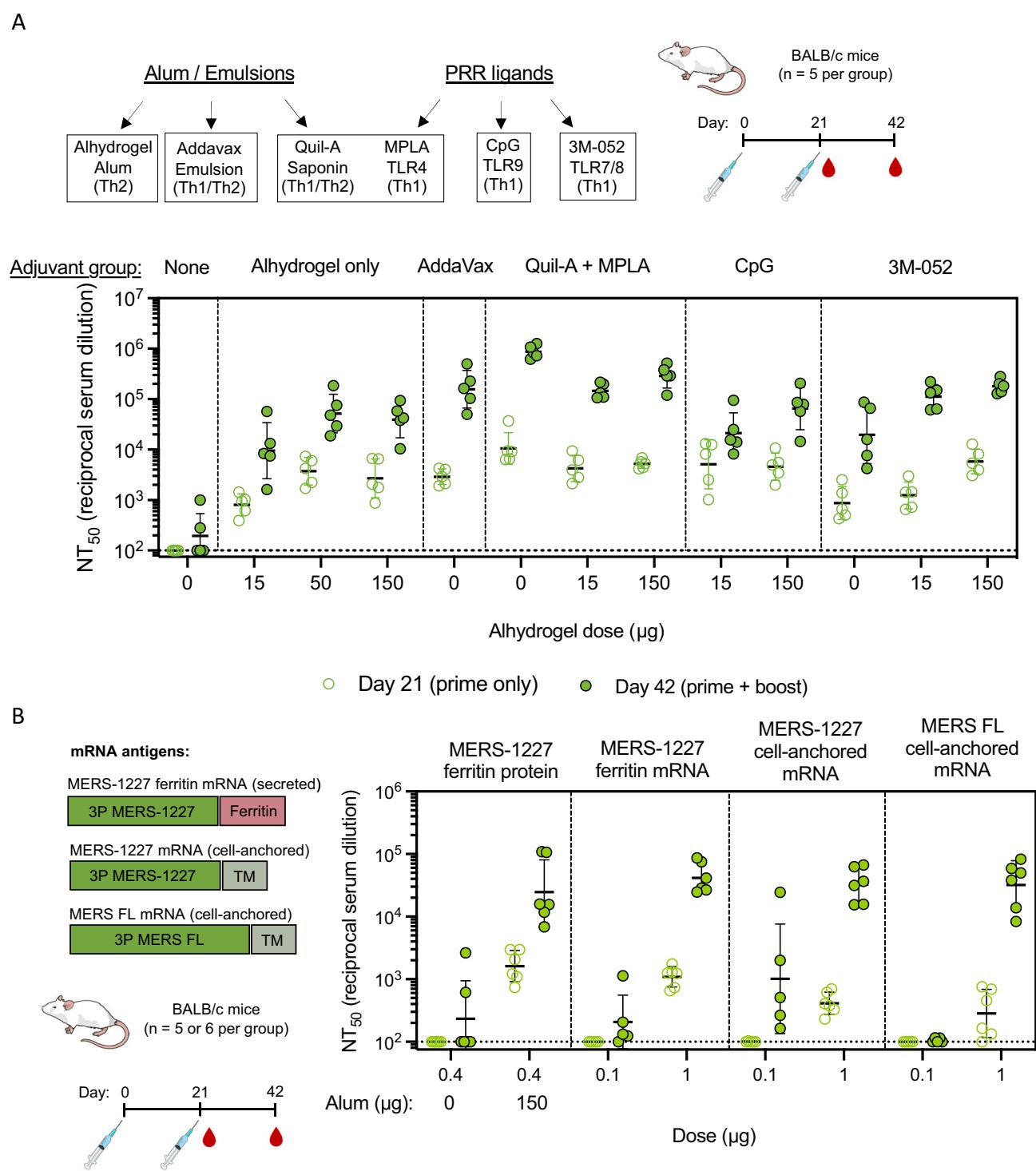

**Fig. 7 | Evaluation of MERS-1227 FNP protein immunized with varied adjuvants and MERS-1227 encoded as an FNP by mRNA reveals optionality for formulation and modality. A** Adjuvants evaluated included a range of alum/emulsion and PRR agonists. Mice ($n$ = 5 per adjuvant group) were immunized at day 0 and 21 with 0.4 µg MERS-1227 FNP protein, and serum was collected at day 21 (open circles) and 42 (closed circles). Each circle represents the NT$_{50}$ value for an individual mouse obtained by performing pseudovirus neutralization analysis in quadruplicate. Lines represent the GMT for each immunization group, and error bars represent the geometric SD. **B** mRNA antigens evaluated for immunogenicity included

mRNA-encoded MERS-1227 ferritin, cell-anchored MERS-CoV spike containing a deletion of ectodomain residues 1228-1294, and cell-anchored MERS-CoV spike with a FL ectodomain. Mice ($n$ = 5 or 6 per group) were immunized at day 0 and 21 with either 0.4 µg MERS-1227 FNP ($n$ = 6), 0.1 µg mRNA ($n$ = 5), or 1 µg mRNA ($n$ = 6). Serum was collected at day 21 (open circles) and 42 (closed circles). Each circle represents the NT$_{50}$ value for an individual mouse obtained by performing pseudovirus neutralization analysis in quadruplicate. Lines represent the GMT, and error bars represent the geometric SD. Source data are provided in the Source Data file. Mouse images from (**A**) and (**B**) from bioart.niaid.nih.gov/bioart/279.

1227 adjuvanted with either 15, 50, or 150 μg Alhydrogel alone elicited an IgG response strongly dominated by IgG1, indicative of a Th2 response, which is well-documented for alum adjuvants[73]. AddaVax was observed to have a slightly decreased Th2-skewing as compared to Alhydrogel alone. Importantly, we observed that co-formulation of Alhydrogel with either Quil-A + MPLA or 3M-052 generated a Th1 bias characterized by a higher IgG2a / IgG1 ratio in these groups compared to Alhydrogel alone. CpG was observed to also overcome the Alhydrogel-induced Th2 skew at the 15 μg Alhydrogel dose, but not at the 150 μg dose. Taken together, the results of this adjuvant screen demonstrate that modification of the adjuvant formulation of MERS-1227 could be used to enhance the resultant antibody responses and potentially modulate the cellular immune response.

### MERS-1227 elicits a comparable response when administered as an adjuvanted protein nanoparticle or an mRNA-encoded ferritin nanoparticle

While protein vaccines exhibit advantages relating to thermostability and lower cost-of-goods, mRNA technology is advancing to meet these challenges and can be produced and modified quickly if needed to respond to a novel or mutated pathogen[74]. For this reason, we evaluated immunization using mRNAs packaged in SM102-containing lipid nanoparticles (LNPs)[75] encoding either MERS-1227 ferritin, a cell-anchored form of the MERS-CoV spike containing a deletion of the ectodomain from residue 1228-1294, or a full-length MERS-CoV spike with no ectodomain deletion (Fig. 7B). All constructs contained 3 P stabilizing mutations, a mutated furin cleavage site, and a 21-residue deletion at the C-terminus of the protein to remove the cytoplasmic tail. We evaluated the biophysical parameters of the mRNA/LNPs including encapsulation efficiency, size, and zeta potential (Fig. S9A) as well as their in vitro cellular potency. To assess cellular potency, we treated HeLa cells with the mRNA/LNP complexes at a series of doses and then stained cells with RBD (JC57-14) or NTD (G2) specific antibodies (Fig. S9B). We compared cell-surface expression (no fixation or permeabilization) with total cellular expression (fixation and permeabilization). As expected, the two cell-anchored constructs (MERS-1227 cell-anchored and MERS-FL cell-anchored) are detected in both cell treatment conditions whereas the MERS-1227 ferritin is only detected when cells are fixed and permeabilized.

We immunized naïve BALB/c mice with either 0.1 μg or 1 μg mRNA on day 0 and day 21 and collected serum at days 21 (prime only) and 42 (prime + boost) (Fig. 7B). As a comparator, we included the MERS-1227 FNP protein +/- 150 μg Alhydrogel. Following a single dose, both the Alhydrogel-adjuvanted MERS-1227 FNP protein and MERS-1227 FNP encoded on mRNA (1 μg dose) elicited similar neutralizing antibody responses to each other and that were higher than those of the two cell-anchored forms of the MERS-CoV spike. After boosting, the 1 μg mRNA groups all exhibited similar levels of pseudovirus neutralizing titers, which were comparable to the adjuvanted MERS-1227 protein nanoparticle. Interestingly, the 0.1 μg mRNA dose groups elicited minimal responses after both doses, and the only responders at this dose level were seen in the MERS-1227 ferritin and the MERS-1227 cell-anchored mRNA groups. This suggests that truncation of the ectodomain to remove the flexible region between 1228 and 1294 conferred a benefit in neutralizing response in the mRNA format, which could provide important insight into the design of other mRNA antigens for vaccine delivery. Furthermore, these results show the versatility of the 3 P stabilized spike we have designed to be administered as a protein nanoparticle, an mRNA-encoded protein nanoparticle, and as a cell-anchored stabilized spike.

### Immunization with MERS-1227 provides dose-dependent protection in hDPP4 BALB/c mice

To evaluate efficacy of MERS-1227 in a small animal model in which we could observe protection from severe disease, we utilized a BALB/c mouse model in which the human DPP4 receptor is expressed in the nasal turbinates, trachea, lungs, and kidney[76]. We immunized hDPP4 transgenic mice (n = 8 per group) with 0, 0.016, or 0.4 μg MERS-1227 formulated with 150 μg Alhydrogel at days 0 and 21 and collected serum at days 21 and 42 (Fig. 8A and Table S3). MERS-CoV pseudovirus and live-virus neutralization titers (Fig. 8B and C) showed dose-dependence, and pseudovirus neutralizing titers were consistent with our evaluation in wild-type BALB/c mice (Fig. S6).

Three weeks following the booster dose, mice were challenged via intranasal inoculation with 10,000 PFU of the EMC/2012 strain of MERS-CoV. Four mice from each group were sacrificed 3 days post-challenge and live-viral titers of MERS-CoV were quantified from the lungs of these animals using plaque assays (Fig. 8D and Table S3). Three mice in the placebo group had viral titers ($\geq 0.9 \times 10^3$ PFU / g lung) in the lung as compared to two mice in the 0.016 μg group and no mice in the 0.4 μg group, demonstrating protection from viral replication in the lungs following MERS-1227 vaccination. The remaining four mice in each group were monitored for weight loss (Fig. S10) and survival (Fig. 8E) until day 10 post-challenge. All mice in the placebo group succumbed to challenge by day 9, whereas 1/4 mice in the 0.016 μg group and 3/4 mice in the 0.4 μg group survived to day 10. Of note, the mouse that died in the 0.4 μg group prior to day 10 was found to be pregnant during necropsy. All mice were necropsied following sacrifice and evaluated for histopathological changes in the lung; minimal changes were observed in both placebo and vaccinated mice as determined by lung scoring metrics (Table S3). Preliminarily, these results suggest that the Alhydrogel-adjuvanted MERS-1227 did not result in vaccine-induced enhanced respiratory disease and lung inflammation following viral challenge. However, future studies involving evaluation of inflammatory infiltrates and characterization of eosinophil and neutrophil content would be necessary to more fully rule out the potential for Th2-type hypersensitivity induced by vaccination prior to challenge[67].

Taken together, the lung viral titer and survival results post viral challenge indicate that immunization with Alhydrogel-adjuvanted MERS-1227 was protective against lethal MERS-CoV challenge in hDPP4 mice in a dose-dependent manner. This provides strong evidence that MERS-1227 is protective and is an important demonstration that clinical development of this stabilized MERS-CoV spike ferritin vaccine could be utilized as a prophylactic countermeasure against MERS.

## Discussion

Emerging pathogens are an ongoing threat to the global population and can be mitigated in part with accessible prophylactic countermeasures. MERS-CoV, a betacoronavirus with substantial morbidity and mortality, circulates in known animal reservoirs and causes sporadic outbreak clusters in humans with a case fatality rate (CFR) of over 30%[7]. Thus, development of a vaccine against MERS is a critical step towards preparedness in the event of a more widespread MERS-CoV outbreak. Here, we designed and evaluated two MERS-CoV vaccine candidates (MERS-1227 and MERS-1236), which display a stabilized form of the MERS-CoV spike on a ferritin nanoparticle. Our efforts primarily focused on a ferritin nanoparticle-based approach, a platform with a favorable safety, reactogenicity, and immunogenicity profile in humans[47,48]. While both MERS-1227 and MERS-1236 FNP proteins exhibited similar immunogenicity in mice, NHPs, and alpacas, MERS-1227 had more favorable structural and stability features as observed by TEM and DSF, and thus we focused our preclinical assessment on this candidate. The stability profile and immunogenicity of MERS-1227 FNP protein indicate that it is a strong candidate for progression into clinical development.

A number of key features differentiate the antigen design we describe here from previous nanoparticle-based MERS vaccine candidates. To our knowledge, we are the first to report an ectodomain

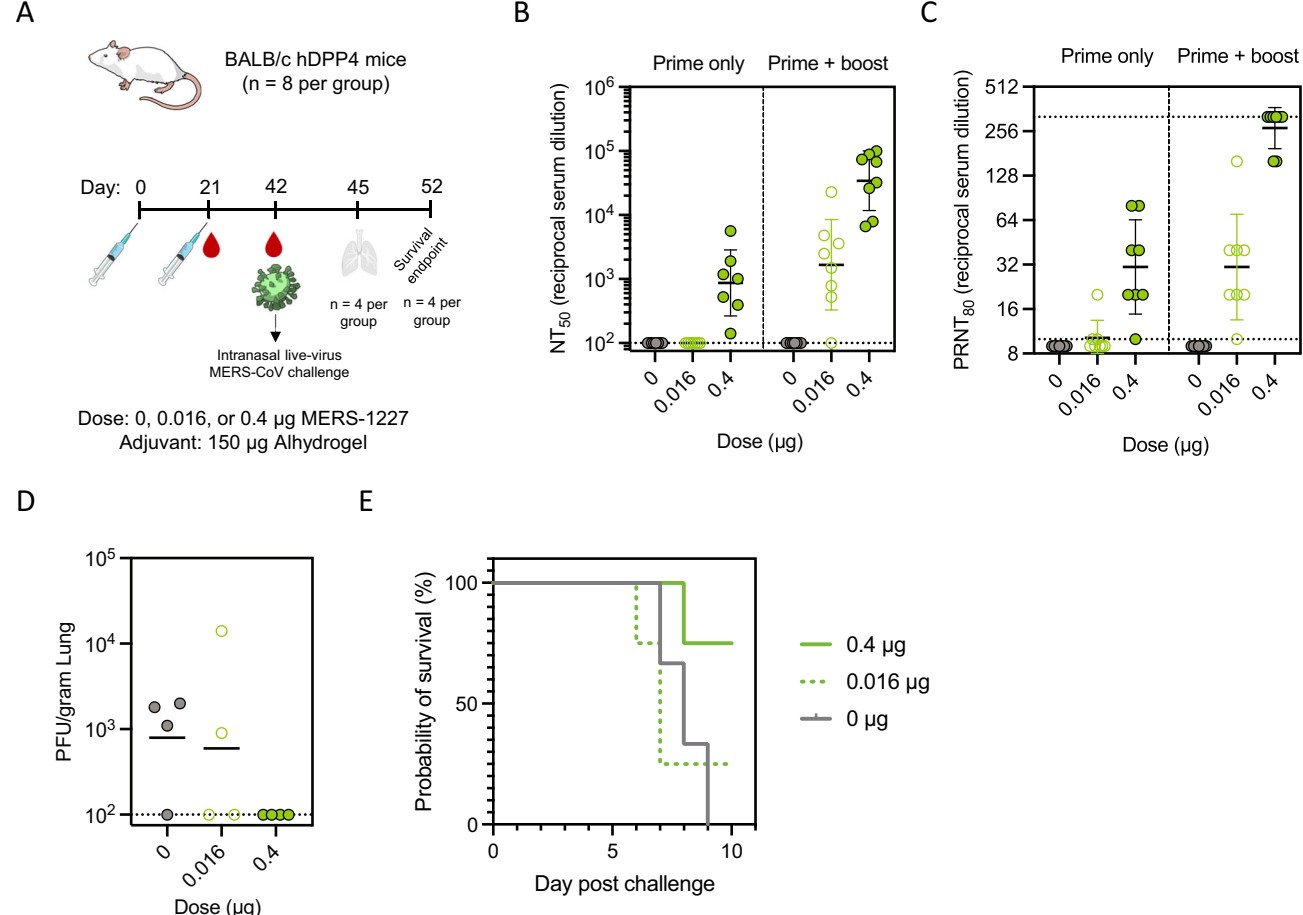

**Fig. 8 | Immunization with MERS-1227 protects human DPP4 transgenic mice from lethal MERS-CoV challenge. A** hDPP4 mouse challenge study design and timeline. Mice were challenged via intranasal inoculation with EMC/2012 MERS-CoV. Each immunization group contained $n = 8$ mice; 4 mice per group were necropsied 3 days post challenge and 4 mice were evaluated for a survival endpoint. Mouse image from bioart.niaid.nih.gov/bioart/279. **B** Pseudovirus neutralizing titers at study day 21 (prime only) and 42 (prime + boost). Each circle represents the $NT_{50}$ value for an individual mouse ($n = 8$ mice per group) obtained by running pseudovirus neutralization analysis in quadruplicate. Serum from one mouse in the 0.016 μg and one mouse in the 0.4 μg dose groups were not analyzed

for day 21 neutralization due to lack of available serum. Lines represent the GMT of the immunization group and error bars represent the geometric SD. **C** Live-virus neutralization obtained from a plaque reduction neutralization assay using day 21 (prime only) and 42 (prime + boost) serum. Each circle represents the $PRNT_{80}$ value for an individual mouse obtained from a single assay run utilizing seropositive and seronegative camel sera as assay controls. Lines represent GMT and error bars represent geometric SD. **D** Quantitation of live MERS-CoV virus in lung samples obtained from mice sacrificed 3 days post-challenge. **E** Survival curves post-challenge. One mouse in the 0 μg group died pre-challenge and is not shown in the survival curve. Source data are provided in the Source Data file.

truncation of the MERS-CoV spike; previously published work described either a full-length spike with the transmembrane domain included[77–79] or a full-length ectodomain conjugated to a nanoparticle[18,80]. The biochemical differences we observe between MERS-1227 and MERS-1236 (Fig. 2C–F) suggest that inclusion of the additional residues of the ectodomain (residues 1237-1294) could lead to enhanced flexibility of the spike, which may have important impacts on nanoparticle stability and manufacturability.

As an indication of the feasibility for scale-up and cGMP manufacturing of MERS-CoV spike ferritin nanoparticles, we showed that both candidates are expressed at high levels (~ 2.7-3.2 g/L titer) in CHO cells (Fig. S1A) and can be purified via two orthogonal tag-less purification processes (Fig. S1B). Furthermore, the unadjuvanted drug substances show a robust thermostability profile in which the molecules remain as folded, stable nanoparticles with retained antigenicity after 14 days at 37 °C (Fig. 3). We also demonstrated nanoparticle stability following low pH treatment, which could be an important process development step required for viral clearance in cGMP manufacturing. These research studies show promise towards feasibility of developing a cGMP chemistry, manufacturing, and controls (CMC) process for these molecules.

The MERS-1227 and MERS-1236 FNPs are immunogenic in multiple animal models (BALB/c mice, NHPs, humanized DPP4 mice, and alpacas). In BALB/c mice, we found that when adjuvanted only with Alhydrogel, MERS-1227 elicits neutralizing antibodies at doses as low as 0.016 μg (Fig. S6). Furthermore, we tested multiple adjuvant combinations that markedly improved the immunogenicity of MERS-1227 (Fig. 7A) demonstrating an opportunity to develop a clinical formulation to elicit the most robust and durable immune response. Finally, a key finding from our mouse immunogenicity studies was that encoding the MERS-1227 ferritin nanoparticle in mRNA elicits a comparable response to immunization with the Alhydrogel-formulated protein nanoparticle (Fig. 7B). Given that protein subunit vaccines and mRNA vaccines have different advantages, it is valuable to observe that MERS-1227 elicits robust immunity after prime and boost injections in multiple platforms.

We evaluated the durability of the immune response to Alhydrogel-adjuvanted MERS-1227 and MERS-1236 following a prime-boost regimen in NHPs with a three-month interval previously defined as an optimal boosting interval in other vaccine regimens[66]. These results revealed that a single dose of either MERS-1227 or MERS-1236 in NHPs elicits neutralizing antibodies that are quantifiable 3 weeks post

vaccination and persist at a steady level until a second dose. The primary response is robustly boosted when a second dose is administered three months after priming (Fig. 5B). Furthermore, antibody levels in vaccinated NHPs were comparable to MERS patient titers following a single dose and exceeded them following a boost when quantified in the same pseudovirus assay (Fig. 5D). In addition to the durable and robust response, serum from immunized NHPs inhibits in vitro infectivity from pseudotyped viruses displaying spike proteins from MERS-CoV clades A, B, C, and a distant pangolin merbecovirus (MjHKU4r-CoV-1) (Fig. 5E). Prior vaccine studies have also shown cross-reactivity to multiple clades[18,81,82], but to our knowledge this is the first preclinical candidate tested against the distant MjHKU4r-CoV-1 strain. This suggests that in the event of crossover of clade C camel strains or potentially a more distant emergent merbecovirus of the HKU4 clade[83] into humans, this vaccine could provide protection.

Additionally, we observed both IFNγ and IL-4 responses to MERS-CoV spike (Fig. 5F), demonstrating that our protein nanoparticle antigens are also effective at eliciting a cellular immune response. The cellular response to MERS-1236 suggested a Th2-biased immune response, whereas the response to MERS-1227 was more balanced. We also conducted an adjuvant screen with MERS-1227 in mice in which the IgG1/IgG2a ratio of the antibody response suggested the ability to modulate the cellular immune response phenotype to these vaccine candidates by changing the adjuvant formulation. This could be important in mitigating the risk for Th2-based immunopathology.

Using an alpaca MERS-CoV challenge model[68], we observed complete protection against viral shedding in animals vaccinated with either 20 or 200 μg of MERS-1227 or MERS-1236 adjuvanted with Alhydrogel. Furthermore, transgenic mice expressing human DPP4[76] were protected from lethal MERS-CoV challenge following immunization with MERS-1227 in a dose-dependent manner, and immunized animals exhibited reduced titers of authentic virus in the lung. This provides strong evidence that these vaccines are effective at protecting against MERS-CoV infection.

Given the robust immune response we observed, including durable neutralizing antibody titers, prevention of infection in an established preclinical challenge model, and a favorable thermostability and manufacturing profile, we assess MERS-1227 as an optimal candidate for rapid clinical development. This would enable prophylactic vaccination for at-risk individuals, thus mitigating the risk of further MERS-CoV outbreaks, and if needed, for regional or global pandemic preparedness, readiness, and response.

## Methods

### Construct design and sequences

An England 1 clade B sequence of MERS-CoV spike protein derived from AFY13307.1[28] was truncated to reflect a similar C-terminal deletion as previously investigated for a SARS-CoV-2 spike[29,45] and was stabilized using proline substitutions at residues 889, 1060, and 1061[25]. The S1/S2 furin cleavage site was mutated from the native RSVR to ASVG[25]. Given the lack of structural resolution at the C-terminal portion of the MERS-CoV spike ectodomain, two truncation positions were tested, one at residue 1227 and one at 1236. An SGG linker was encoded following the truncation site prior to a modified *H. pylori* ferritin as previously described[29]. The terminal 3 residues in MERS-1227 encode a potential glycosylation site (NST). An N1225G mutation was made to remove this glycosylation site, as it was hypothesized that a glycan at the base of the trimer immediately preceding the ferritin component could impede nanoparticle formation.

### MERS-1227 and MERS-1236 stable CHO cell pool generation, fed-batch culture production, and purification

Stable CHO cell pools were generated as previously described[49]. Briefly, codon optimized coding sequences were synthesized and cloned into Leap-In™ transposon-based expression constructs. The plasmids were co-transfected with Leap-In™ transposase mRNA into miCHO-GS host cells (ATUM) and the stable pools were established by glutamine-free selection. For fed-batch productions, cells were seeded at $0.75 \times 10^6$ per mL in shake flasks at either 25 or 100 mL scale grown at 37 °C with 5% $CO_2$. Growth temperature was adjusted to 30 °C on day 4 and cells were supplemented throughout the fed-batch production using proprietary feed solutions. Cells were harvested on day 14 by spinning at 7000 x g for 10–20 min. Supernatants were filtered using a 0.22 μm filter.

For research-scale purifications, supernatants were diluted with 20 mM Tris pH 8.0, 1 M sodium chloride to a final concentration of 200 mM sodium chloride (not accounting for sodium chloride in the cell culture media). Diluted supernatants were then passed over a Cytiva 5 mL HiTrap Q anion exchange column equilibrated in 20 mM Tris pH 8.0 using an AKTA fast protein liquid chromatography (FPLC) system run with UNICORN software (version 7.8.0.2159). MERS-CoV ferritin nanoparticles were collected from the column flow-through and then dialyzed overnight at 4 °C using cellulose ester 1 MDa molecular weight cutoff dialysis tubing (Spectrum) into 20 mM Tris, pH 7.5, 150 mM sodium chloride. Dialyzed material was concentrated to a final volume of ~0.5-2 mL using an Amicon spin concentrator (100 kDa molecular weight cutoff). Concentrated samples were then filtered using a 0.22-μm filter and injected onto a Cytiva Superose 6 10/300 GL (PN 29091596) or Sepax 10 x 300 mm SRT SEC-1000 (particle size 5 μm, pore size 1000 Å, PN 215950-10030) size-exclusion chromatography column equilibrated in 20 mM Tris pH 7.5, 150 mM sodium chloride using an AKTA FPLC. Protein-containing fractions were assessed by SDS-PAGE analysis, and fractions containing homogenous protein were pooled. Fractions were supplemented with 5% sucrose as a cryoprotectant, frozen in liquid nitrogen, and stored at −80 °C until use. The formulation buffer used throughout the described studies was 20 mM Tris pH 7.5, 150 mM sodium chloride, 5% sucrose.

Alternatively, for bind and elute purifications (Fig. S1B), MERS-1227 14-day fed-batch supernatant was treated with Pierce Universal Nuclease and then diluted 5-fold into 20 mM Tris pH 7.5. Diluted supernatant was injected onto a 1 mL Sartobind Q Nano (Sartorius No.96IEXQ42DN-11--A) equilibrated in 20 mM Tris-Cl pH 7.5 at 5 mL per min. FNP was eluted using a step elution with 10 column volumes of 14, 30, and 100% 20 mM Tris-Cl pH 7.5, 1 M sodium chloride. FNP-containing fractions were pooled, brought to 1.6 M ammonium sulfate, and loaded onto a 1 mL Cytiva HiTrap Hydrophobic Interaction Capto butyl column equilibrated in 1.8 M ammonium sulfate in 50 mM HEPES pH 7.5. The protein was eluted with a gradient of 50 mM HEPES pH 7.5.

### SDS-PAGE

Denaturing SDS-PAGE analysis was performed by diluting proteins with LDS sample buffer (containing 2-mercaptoethanol for reducing conditions) and heating samples for 5 min at 95 °C. Samples were then loaded on a 4-20% MiniPROTEAN TGX precast gel and run for 30 min at 230 V. Gels were stained using GelCode Blue staining reagent and imaged using a Thermo Fisher Scientific imager (iBright CL750 1.8.0).

### Analytical size-exclusion chromatography multi-angle light scattering

Purified MERS-1227 and MERS-1236 were diluted to 0.125–0.5 mg/mL in 20 mM Tris, pH 7.5, 150 mM sodium chloride, 5% sucrose. For each run, 5–10 μg protein was loaded onto a 4.6 x 300 mm SRT SEC-1000 analytical column (particle size 5 μm, pore size 1000 Å, PN 215950-4630) pre-equilibrated in 20 mM Tris pH 7.5, 150 mM sodium chloride run at 0.35 mL/min on a Waters Arc Premier high-pressure liquid chromatography system run with Empower software (version 7.3.0). Multi-angle light scattering was measured using a Wyatt DAWN and refractive index was measured using a Wyatt Optilab run with ASTRA software (version 8.3.0.132). Molecular weight calculations were performed

using Wyatt ASTRA software (version 8.3.0.132) by utilizing the light scattering and refractive index signals from each sample.

## Sample preparation for LC-MS/MS peptide mapping

MERS-1227 and -1236 FNP proteins were proteolytically digested using S-Trap columns (Protifi, C02-micro-10) following the manufacturer protocol with minor modifications. All digestions were performed in triplicate. For each replicate, 25 μg protein in formulation buffer was added to an equal volume of 2x lysis buffer (5% SDS, 100 mM Tris, pH 8.5). Proteins were reduced in 5 mM dithiothreitol (DTT) for 30 min at 55 °C and alkylated in 20 mM iodoacetamide for 30 min in the dark at room temperature. Phosphoric acid was added to a final concentration of 2.5% (v/v). Acidified samples were diluted in 10 volumes of 100 mM Tris, pH 7.55 in 90% methanol/10% water and applied to S-Trap columns by centrifugation at 4000 x $g$. The column was washed in the same buffer. 2.5 μg of trypsin (Promega, #V5113) diluted in 20 μL 50 mM ammonium bicarbonate, pH 8 was applied to the column and allowed to sit overnight at 37 °C. Peptides were eluted in 40 μL ammonium bicarbonate pH 8, followed by 40 μL 0.2% formic acid in water, and finally 40 μL 50% acetonitrile in water. Eluted peptides were dried on a SpeedVac and resuspended in 30 μL 0.1% formic acid in water prior to analysis.

## LC-MS/MS peptide mapping method

Peptides were analyzed by LC-MS/MS using a Vanquish Horizon LC directly interfaced to an Orbitrap Exploris 240 mass spectrometer (Thermo Fisher Scientific) run with Xcalibur software (version 4.7). 6 μL of resuspended peptides were separated by reversed-phase LC on a Accucore Vanquish C18+ (100 mm length x 2.1 mm ID, 1.5 μm particle size) at 0.3 mL/min using a linear gradient of 5–30% mobile phase B (0.1% formic acid in acetonitrile) over 25 min followed by a ramp to 90% mobile phase B over 3 min, where mobile phase A was 0.1% formic acid in water. Positive electrospray ionization was achieved with a spray voltage of 3500 V, a sheath gas flow rate of 40 arbitrary units, auxiliary gas heated to 250 °C at a flow rate of 15 arbitrary units, and a sweep gas flow rate of 1 arbitrary unit. The ion tube transfer temperature was held at 320 °C and the RF lens was set to 70. MS analysis was carried out in a Top 10 data-dependent acquisition mode. Full MS scans were acquired at a resolution of 60,000 (FWHM @ m/z 200) with an automatic gain control target of "standard", m/z range of 350–1800, and maximum ion accumulation time of "auto". Precursor ions with charges 2–6 and a minimum intensity threshold of 1 x 10⁴ were selected with a quadrupole isolation window of 2.0 m/z for HCD fragmentation with a normalized collision energy (NCE) of 27%. MS/MS scans at a resolution of 15,000 (FWHM @ m/z 200) were acquired with an automatic gain control of "standard", a maximum ion accumulation time of 150 ms, and a fixed first mass of 120 m/z. Precursors selected for fragmentation more than 2 times within a 15 second window were dynamically excluded from additional fragmentation for 10 seconds.

## LC-MS/MS peptide mapping data analysis

To identify post-translational modifications of MERS-1227 and -1236 ferritin nanoparticles, raw mass spectrometry data were analyzed using the GlycanFinder module of PEAKS Studio 11 (Bioinformatics Solutions, Inc.). Custom FASTA databases were generated comprising the Chinese hamster (*Cricetulus griseus*) proteome (downloaded on November 27, 2023), common mass spectrometry contaminant proteins[84], and either the MERS-1227 or -1236 FNP protein sequence. The raw MS data were searched against the database with the following modifications: fixed carbamidomethylation on Cys, variable oxidation of Met, and variable deamidation of Gln and Asn, with a maximum of two variable modifications per peptide. Glycans were identified using the default N-linked and O-linked databases within PEAKS GlycanFinder. Semi-specific tryptic peptides with a maximum of two missed cleavages were considered. The allowed mass tolerances were

10 ppm for precursor ions, 0.04 Da for peptide product ions, and 20 ppm for glycan diagnostic ions. Hits were filtered to a false discovery rate of 1% using the PEAKS decoy-fusion approach. For estimation of the relative quantities of the glycan compositions at each glycosite, the average integrated precursor peak areas for each detected glycan composition were calculated and reported as the percentage of the total glycan peak areas at a given site. Glycans with quantifiable peak areas in at least 2/3 of the experimental replicates were considered for relative quantitation. Data were analyzed and plotted using ggplot2 (version 3.5.0) and R (version 4.3.3).

## Dynamic light scattering and differential scanning fluorimetry

Proteins were diluted to 0.125 - 0.5 mg/mL in formulation buffer (20 mM Tris pH 7.5, 150 mM sodium chloride, 5% sucrose) and loaded into glass capillaries (NanoTemper). Samples were then analyzed on a Nanotemper Prometheus Panta run with PR.Panta Control software (version 1.9) using DLS with a viscosity parameter of 1.149 mPa·s and a refractive index parameter of 1.341 to determine the cumulant radius of the particles and DSF to determine the melting temperature ($T_m$). Melting temperature runs were performed using a gradient of 1.0 °C per minute from 25 °C to 90 °C. Data was processed using PR.Panta Analysis software (version 1.9).

## Transmission electron microscopy

For TEM analysis, purified MERS-1227 and MERS-1236 were diluted using 20 mM Tris pH 7.5, 150 mM sodium chloride. A 3.5 μL drop of diluted sample suspension was applied to a TEM grid overlaid with a 3-4 nm layer of amorphous C (CF200-CU-UL, Electron Microscopy Sciences). Prior to sample application, the carbon-coated grid was plasma-cleaned for 15 seconds via a lab-made device. The 3.5 μL of sample was allowed to incubate on the carbon surface for one minute. After blotting the sample away with filter paper, each grid was twice dipped quickly (~1 second) into separate drops of de-ionized water followed by blotting with filter paper. This double washing and blotting was then repeated with two drops of 1% ammonium molybdate. For all steps, the next dipping was performed before the grid could completely dry. Finally, each grid was dipped into a drop of 1% ammonium molybdate solution. After 15-20 seconds, the stain was blotted away with filter paper, and the grids were allowed to air-dry. To collect hundreds of images rapidly, specimens were imaged on a ThermoFisher Titan Krios transmission electron microscope equipped with a Gatan Bioquantum K3 energy filter and direct electron detector. The microscope was operated at 300 kV and at liquid nitrogen temperatures, and the program SerialEM (version 4.2.4) was used to collect images[85]. Images were collected as multiple frames of the same field of view.

After imaging, micrographs were processed to give two-dimensional class averages via the RELION software package (version 5.0.0)[86]. Frames in the multi-framed images were aligned and summed via the MOTIONCOR2 algorithm[87] implemented in RELION. Contract transfer function parameters were determined for each summed micrograph via CTFFIND4[88], via an interface in RELION. Several hundred particle images were selected manually and used to train the program Topaz (interfaced in RELION) to automatically pick thousands more particle images[89]. Extracted particle images were then aligned in two dimensions to give 2D class averages via the general Bayesian approach of RELION[90].

## Biolayer interferometry

Biolayer interferometry analysis of MERS-1227 and MERS-1236 was performed using an Octet R8 instrument run with Octet BLI Discovery software (version 13.0.1.19) with anti-hIgG Fc capture (AHC) biosensors. Antibodies were procured from GenScript with human Fc domains and stored in 1X TBS pH 7.4. $V_H$ and $V_L$ sequences are found in Supplementary Data 2. Antibodies were diluted to 50 μg/mL and MERS-

CoV nanoparticles were diluted to 100 μg/mL in 1X Sartorius kinetics buffer (PBS + 0.1% BSA, 0.02% Tween-20, and a microbicide, Kathon) and pipetted into black-walled, black-bottom plates (200 μL per well). AHC tips were dipped into antibody wells and subsequently dipped into nanoparticle wells to assess binding association for 120 s. Dissociation was monitored for 120 s by moving tips into a well containing buffer alone. Data was analyzed using Octet Analysis Studio software (version 13.0.1.35).

### pH and temperature stability treatment of MERS-CoV FNPs

To assess the pH stability of MERS-CoV FNP proteins, aliquots of the formulation buffer (20 mM Tris pH 7.5, 150 mM NaCl, 5% sucrose) were adjusted to pHs of 2, 3, 3.5, and 4 (+/- 0.1 pH unit) by dropwise addition of hydrochloric acid. Slide-A-Lyzer MINI Dialysis Devices (15 mL conical format, Thermo Fisher Scientific Cat #88401) were briefly rinsed in these buffer solutions. 350 μL of MERS-1227 or MERS-1236 FNP proteins at 0.25 mg/mL in formulation buffer were added to each dialysis device and dialyzed against 14.5 mL of the pH-adjusted formulation buffers (or pH 7.5 as a control). Proteins were dialyzed at room temperature for 1 h with gentle agitation. After 1 h incubation, 1 μL dialyzed protein was used to confirm the target pH of the solution using pH paper. At this time, dialysis solution was discarded and replaced with a fresh 14.5 mL pH-adjusted formulation buffer and dialysis was allowed to proceed for an additional 1.5 h. Proteins were then dialyzed back to pH 7.5 by replacing dialysis solution with pH 7.5 formulation buffer followed by a 1 h incubation at room temperature. pH 7.5 formulation buffer was then replaced with fresh pH 7.5 buffer and dialysis was allowed to proceed overnight at 4 °C. Proteins were then removed from dialysis devices, transferred to 2 mL polypropylene tubes, snap frozen using liquid nitrogen, and stored at -80 °C until analysis. For analysis, all samples were thawed at the same time and diluted to 0.125 mg/mL with formulation buffer and evaluated using analytical SEC, DLS, DSF, and BLI as described above.

To assess the thermal stability of MERS-CoV FNP proteins, purified proteins were diluted to 0.5 mg/mL in 20 mM Tris, pH 7.5, 150 mM NaCl, 5% sucrose, and aliquoted in 2 mL polypropylene tubes. Samples were placed at either 4, 22 (ambient temperature), or 37 °C. Control samples were snap-frozen immediately following dilution to 0.5 mg/mL and 90 °C heat-treated controls were heated to 90 °C for 30 min and then snap-frozen and stored at -80 °C until analysis. Aliquots of 4, 22, and 37 °C were taken following 7- and 14-day incubations and snap-frozen at each time-point until analysis. For analysis, all samples were thawed at the same time and assessed at 0.5 mg/mL using analytical SEC, DLS, DSF, and BLI as described above.

### Antigen production and purification

MERS-CoV GCN4 trimer, MERS-CoV RBD, and MERS-CoV NTD (amino acid sequences in Supplementary Data 2) were produced in Expi293F cells and purified using His-tag purification. WT *H. pylori* ferritin was expressed in Expi293F cells and purified as previously described[29]. Expi293F cells were cultured in Expi293 media (Gibco) at 37 °C, 120 RPM, and 5% $CO_2$. For transfection, cells were grown to a cell density of 3 x 10⁶ cells per mL and transfected with plasmid DNA at 1 μg DNA per mL cell culture using ExpiFectamine293 transfection reagent (Gibco) according to manufacturer recommendations. Cells were harvested 4−6 days post transfection by centrifugation at 7000 x g followed by filtration using a 0.22-μm filter.

Filtered supernatant was diluted with 10X PBS to a final concentration of 2.5X PBS (pH 7.4) and loaded onto either a HisTrap excel (Cytiva) or a HiTrap TALON crude (Cytiva) column pre-equilibrated in 2.5X PBS pH 7.4. Proteins were eluted with 250 mM imidazole in 2.5X PBS pH 7.4. For Luminex, MERS-CoV GCN4 trimer was buffer exchanged into 20 mM Tris pH 7.5, 150 mM NaCl and biotinylated using a BirA biotinylation kit (Avidity). MERS-CoV RBD and NTD were purified on a Cytiva Superdex 200 Increase 100/300GL (PN 28990944) SEC column

pre-equilibrated in 1X PBS pH 7.4. MERS-CoV GCN4 trimer was purified on a Cytiva S6 10/300 GL (PN 29091596) or a Cytiva Superdex 200 Increase 10/300 GL (PN 28990944) SEC column pre-equilibrated in 20 mM Tris pH 7.5, 150 mM NaCl. Protein-containing fractions were pooled, snap-frozen, and stored at -80 °C until use.

### MERS-CoV FNP formulation for immunization studies

To formulate protein antigens with Alhydrogel adjuvant, the antigens were first diluted to the required concentration in formulation buffer (20 mM Tris pH 7.5, 150 mM NaCl, 5% sucrose). Alhydrogel adjuvant (Invivogen #vac-alu-10, 10 mg/mL) was resuspended by inverting 50 times, and then the required volume was added to the diluted antigen. The combined antigen-adjuvant mixture was then inverted 50 times, maintained at room temperature for 30-60 min to allow for antigen binding to Alhydrogel, and then inverted 50 additional times.

For the adjuvant screen (Fig. 7A), MERS-1227 FNP was first formulated with Alhydrogel adjuvant as described above, except the FNP and Alhydrogel concentration were initially prepared to 2x the required final concentration for injection. For formulations without Alhydrogel, only the protein was included. Additional adjuvants were prepared as follows: ODN2395 (CpG, Invivogen Cat #trtl-2395) was dissolved to 1 mg/mL in sterile water, MPLA-SM (Invivogen Cat. #vac-mpla2) was dissolved to 1 mg/mL in sterile DMSO, and QuilA was dissolved to 1 mg/mL in sterile water. AddaVax (InvivoGen Cat t#vac-adx-10) was purchased as a 2x ready-to-use suspension. 3M-052-AF was provided at 0.050 mg/mL by The Access to Advanced Health Institute (AAHI). For each formulation, the appropriate volume of adjuvant plus additional formulation buffer was added to bring the final concentration to that required for injection, and the formulated vaccines were inverted 50 additional times.

### Mouse immunization studies

Female BALB/c mice (*Mus musculus*), age 7-8 weeks, were purchased from Charles River Laboratories and acclimated for at least one week in a contract vivarium (Fortis Life Sciences) prior to immunization. Antigens and adjuvants were formulated to deliver the indicated doses in 100 μL volume. Mice were injected intramuscularly with a 50 μL dose of antigen in each hind leg to deliver 100 μL of vaccine formulation per immunization. To obtain blood via retroorbital bleeding, a capillary tube was inserted into the medial canthus. Blood was then transferred to an Eppendorf or BD Microtainer Blood Collection Tube (BD 365967), allowed to clot at room temperature for approximately 2 h, and then spun down at 4000 rpm in a tabletop centrifuge. The serum layer was transferred to an Eppendorf tube and frozen at −80 °C. Prior to subsequent assays, serum was heat inactivated at 56 °C for 30 min.

### Luminex bead conjugation, total IgG binding, and IgG subclass binding assays

To measure the total IgG response and IgG subclasses in mice and NHPs, we used Luminex assays adapted from a previously described method[91]. Biotinylated MERS-CoV GCN4 trimer (50 μg) was incubated with 2 x 10⁶ MagPlex®-Avidin Microspheres (Luminex #MA-A012-01) for 2 h, then washed and resuspended in 1 mL Luminex wash buffer (0.1% BSA, 0.02% Tween-20 in PBS pH 7.4). For other antigens (MERS-CoV RBD and MERS-CoV NTD), different MagPlex® Microspheres (Luminex #MC10026-01 to MC10029-01 and MC10034-01) were used. Activation involved 1 x 10⁷ microspheres with 50 mg/mL sulfo-NHS (Thermo Fisher Scientific #24510) and EDC (Thermo Fisher Scientific #22980) for 20 min, followed by adding 50 μg of antigen for a 2-h incubation. Conjugated microspheres were resuspended in 1 mL Luminex wash buffer. The post-conjugation antigenicity of each antigen was validated by measuring binding to control mAbs diluted to 20 μg/mL: 3A3 (MERS-CoV GCN4 trimer S2)[57], JC57-14 (RBD)[13], and G2 (NTD)[54].

Serum samples from mice and NHPs were diluted 50-fold in Luminex diluent (1% non-fat milk, 5% FBS, 0.05% Tween-20 in PBS pH 7.4). 25 µL of diluted serum was added to 25 µL of antigen-conjugated microsphere suspension (at least 1000 microspheres / antigen). After a 30-min incubation, microspheres were washed three times with Luminex wash buffer. For total IgG detection, microspheres were incubated with 100 µL of goat anti-mouse IgG R-phycoerythrin (2 µg/mL, SouthernBiotech #1030-09) for 30 min. For mouse IgG subclasses, biotinylated secondary antibodies specific to IgG1 and IgG2a (4 µg/mL, SouthernBiotech #1070-08, #1080-08) were used, followed by 30 min with streptavidin-PE (5 µg/mL, Thermo Scientific #12-4317-87). Results were read on a Luminex xMAP Intelliflex system using INTELLIFLEX software (version 2.1.1015). For Fig. 4B, statistical analysis was performed in GraphPad Prism (version 10.5.0) using a two-way ANOVA with Tukey's multiple comparisons.

### MERS-CoV spike-pseudotyped lentivirus production and viral neutralization assay

To assess elicitation of neutralizing antibodies by MERS-1227 and MERS-1236, we utilized a MERS-CoV spike-pseudotyped lentivirus neutralization assay. For virus production, HEK293T (ATCC CRL-3216) cells were cultured in cell-growth media (DMEM, 10% fetal bovine serum, 2 mM L-glutamine, 100 Units/mL penicillin, and 100 µg/mL streptomycin) and seeded at a density of 6 million cells in a 10-cm dish. One day following cell seeding, cells were transfected with the following plasmids: 3.4 µg MERS-CoV spike plasmid, 2.2 µg of each lentivirus helper plasmid (pHDM-Hgpm2, pHDM-tat1b, and pRC-CMV-rev1b)[60], and 10 µg lentivirus packaging vector (pHAGE-CMV-Luc2-IRES-ZsGreen-W)[60] using BioT transfection reagent. Lentivirus production plasmids were obtained through BEI Resources (NR-52948). For all pseudovirus neutralization panels with the exception of Fig. 5C, the MERS-CoV spike plasmid encoded the spike protein (EMC/2012 strain) with the µ-phosphatase signal peptide (sequence in Supplementary Data 2) and an 18-amino acid C-terminal truncation, which has previously been shown to improve viral titers. For viruses used to generate the panel shown in Fig. 5D, native sequences of all viral spike proteins were used, and an 18 amino acid C-terminal truncation was also included (sequences in Supplementary Data 2). The media was removed and replaced with fresh media 18-24 h post transfection. Viruses were harvested 72 h post transfection by spinning at 300 xg for 5 min and filtering through a 0.45-µm PES filter. Viruses were frozen at -80 °C until use.

Infectivity assays with MERS-CoV spike-pseudotyped lentiviruses were performed in HeLa cells expressing human DPP4 (Cellecta RMCOV-CDPP4HE). HeLa/DPP4 cells were plated in white-walled, white-bottom 96-well plates at 5,000 cells per well one day prior to or the day of infection. To quantify neutralizing antibody titers in serum from immunized mice, serum samples were diluted in cell-growth media. Serum dilutions were then mixed with MERS-CoV spike-pseudotyped lentivirus and virus/serum dilutions were incubated at 37 °C for 1 h. Media was either aspirated off of HeLa/DPP4 cells and replaced with virus/serum dilutions, or trypsinized HeLa/DPP4 cells were plated directly into virus/serum dilutions and left to incubate with cells for 72 h at 37 °C. Infectivity in each well was quantified using BriteLite and luciferase signal was read out using a BioTek Synergy H1 microplate reader using Gen5 software (version 3.13.15). Data was analyzed using GraphPad Prism (version 10.5.0). Control wells containing virus only and cells only were averaged, and serum dilution infectivity values were normalized to these averages set at 100% infectivity (virus only) and 0% infectivity (cells only). Normalized values were then fit with a 3-parameter non-linear regression (Y=Bottom + (Top-Bottom)/(1 + (X/IC50)). For Fig. 4C, statistical analysis was performed in GraphPad Prism using a two-way ANOVA with Tukey's multiple comparisons.

### NHP immunization studies

Ten male cynomolgus macaques of Mauritian origin (*Macaca fascicularis*), 6 years of age, were used for immunization studies (Table S2). Animals were maintained at Alpha Genesis, Inc. Alhydrogel-adjuvanted MERS-1227 and -1236 were prepared to a final concentration of 0.1 mg/mL antigen and 1.5 mg/mL Alhydrogel adjuvant. NHPs were injected intramuscularly with 500 µL of vaccine formulation per immunization. To obtain blood, animals were sedated with ketamine HCL 10-20 mg/kg IM and blood was collected from the femoral vein using a 22 G 1.5-inch needle, vacutainer sheath, and collection tube. Blood was allowed to clot at room temperature for approximately 1 h and was then centrifuged at 2000 xg for 20 min. Serum was transferred to Eppendorf tubes, heat inactivated at 56 °C for 30 min, and frozen at -80 °C.

For the collection of PBMCs, blood was carefully layered on top of a Ficoll Paque and then centrifuged at 2,000 xg for 20 min at 21 °C. Plasma was pipetted off into a separate tube. The mononuclear cell layer (buffy coat) was pipetted into conical tubes containing wash buffer (Hanks Balanced Salt Solution + 2% Fetal Bovine Serum) and then filled with wash buffer and mixed by inversion. Samples were then centrifuged at 400 xg for 7 min at 21 °C. Wash buffer was then aspirated off and ACK Lysis Buffer was added to lyse red blood cells. Samples were incubated in lysis buffer for 3 min, then tubes were filled with wash buffer to dilute lysis buffer and centrifuged at 400 xg for 5 min at 21 °C. Wash buffer was aspirated off and cell pellets were resuspended in wash buffer. Cells were centrifuged at 400 x g for 5 min at 21 °C and resuspended in freezing buffer (wash buffer + 10% DMSO) and stored in $LN_2$ until use.

### VSV pseudovirus production and neutralization assay

VSV pseudotyped with the full-length MERS-CoV EMC/2012 spike protein[92] was produced as previously described[14,18,93]. In brief, 5 x 10^6 HEK293T cells were split into poly-D-lysine-coated 10 cm² plates and incubated overnight at 37 °C and 5% $CO_2$ until they reached 80-90% confluency. The cells were then transfected with the construct encoding the MERS-CoV spike protein using Lipofectamine 2000 (Invitrogen) following manufacturer's recommendations. The cells were incubated at 37 °C and 5% $CO_2$ for 20–24 h after which the cells were washed three times with DMEM (Gibco) and infected with VSVΔG/luc[94] for 2 h at 37 °C and 5% $CO_2$. Next, the cells were washed five times with DMEM and incubated in DMEM supplemented with an anti-VSV-G antibody (I1-mouse hybridoma supernatant diluted 1:25, from CRL-2700, ATCC) at 37 °C and 5% $CO_2$. After 20–24 h the supernatant was harvested, clarified by centrifugation, filtered using a 0.45-µm filter, and concentrated with a 100 kDa MWCO centrifugal filter (Amicon). The resulting pseudovirus was frozen at -80 °C until use.

Neutralization assays using VSV pseudotyped with the MERS-CoV EMC/2012 spike were performed as previously described[14,18,93]. Briefly, Vero-TMPRSS2 cells[95] were split into white-walled, clear bottom 96-well plates at a density of 22,000 cells per well and cultured overnight at 37 °C and 5% $CO_2$ until they reached approximately 90-95% confluency. Serum and plasma samples were diluted in DMEM starting at 1:66 (NHP samples) or 1:10 (MERS patient samples) and serially diluted 1:3 thereafter. The diluted sera or plasma were then mixed with pseudovirus diluted 1:50 in DMEM and incubated at room temperature for 45 min. Vero-TMPRSS2 cells were washed three times with DMEM and subsequently the sera/plasma-pseudovirus mixtures were transferred to the cells. The cells were incubated for 2 h at 37 °C and 5% $CO_2$ after which an equal volume of DMEM supplemented with 20% FBS (Gibco) and 2% Pen-Strep (Gibco) was added to each well. After 17-20 h, ONE-Glo EX substrate (Promega) was added to each well and the plates were incubated for 5-10 min in the dark at 37 °C. Luminescence values were recorded using a BioTek Synergy *Neo2* plate reader. The data were normalized in GraphPad Prism 10 using the relative light units (RLU) values obtained for uninfected cells to define 0% entry and the

RLU values measured for cells infected with pseudovirus only to define 100% entry. NT50 values were determined from the normalized data using a [inhibitor] vs. normalized response – variable slope model using two technical replicates to generate the curves. Plotted NT50 values were obtained by averaging the NT50 values obtained from the two independent biological replicates using two distinct batches of pseudovirus.

### IFN-γ and IL-4 ELISpot assays

ELISpot plates (Millipore MSIPS4W10) were activated with 50% ethanol, washed with water, and coated with either IFN-γ or IL-4 monoclonal antibody (Mabtech 3420-2 A and 3410-2 A) at 15 μg/mL in PBS overnight at 4 °C. Plates were washed with PBS and blocked with complete RPMI for 1 h at 37 °C. Blocking reagent was removed, and 50 μL protein (MERS-CoV spike or *H. pylori* ferritin) at 20μg/mL was added to the corresponding wells. PBMCs were then added to wells in a volume of 50μL at a concentration of $4 \times 10^6$ cells/mL, resulting in $2 \times 10^5$ cells per well at a final protein concentration of 10 μg/mL. Media and phytohemagglutinin were used for quantitative and qualitative controls, respectively. The plates were placed in a 37 °C incubator with 5% $CO_2$ for 16–20 h. After washing, anti-human IFN-γ biotin or IL-4 biotin detector antibody was added to each well at 1 μg/mL in PBS pH 7.4 + 0.5% BSA. The plates were incubated for approximately 2 h at room temperature. After washing, Streptavidin-ALP was added to each well at 1:1000 in PBS pH 7.4 + 0.5% BSA and plates were incubated for 1 h at room temperature. The plates were washed with PBS and NBT/BCIP chromogen (Thermo Scientific 34042) was added to each well and incubated at room temperature for 10 min. The chromogen was removed and the plate was rinsed thoroughly with tap water. Plates were then dried and removed from direct light for 24 h. Spots were counted by ZellNet using a Zeiss KS ELISpot reader using KS ELISPOT software (version 4.10). The median of duplicate well cell counts was calculated, and background subtraction was performed per individual monkey by subtracting the median spot-forming unit (SFU) value of medium-stimulated wells from each median SFU value of the protein-stimulated sample. Values were multiplied by 5 to yield SFU/$10^6$ PBMC.

### Alpaca challenge study

Twenty-five female alpacas (*Vicugna pacos*), 5 to 9 years of age, were assigned randomly such that their ages were homogeneously distributed to 5 vaccine groups (Table S2) and identified by thermally sensitive microchip. Alpacas were immunized at study days 0 and 21 by intramuscular injection of Alhydrogel-adjuvanted nanoparticle or Alhydrogel only (placebo group). Two days prior to the challenge on day 42, alpacas were moved into a large animal BSL3 facility. Blood was collected at the time of each immunization and on day 41, and sera were stored frozen until use.

Challenge was conducted as described previously[68]. Briefly, the alpacas were sedated with xylazine and challenged by intranasal instillation of a total of $5.4 \times 10^5$ plaque-forming units (back titrated dose) of the EMC/2012 strain of MERS-CoV, split into 1 mL per nare. Alpacas were evaluated clinically daily for 7 days post-challenge. A deeply inserted nasal swab from each side was collected daily on days 1 to 5 and 7 post-challenge. Swabs were broken off in tubes containing 1 mL of BA1 (Tris-buffered MEM containing 1% bovine albumin) supplemented with 5% fetal bovine serum and antibiotics (gentamycin, penicillin, and streptomycin), then stored frozen until assay. Alpacas were euthanized between 7 and 11 days post-challenge. Virus in nasal swab samples was titrated using a double overlay plaque assay on Vero cells (ATCC CCL-81)[61]. Titers of neutralizing antibodies to MERS-CoV in sera were determined by plaque reduction neutralization[96] and expressed as the reciprocal of the highest dilution resulting in 90, 80, or 50% neutralization of virus. Sera were tested at two-fold dilutions from 1:10 to 1:320.

### Design and procurement of MERS-1227 mRNA/LNP complexes for immunization

mRNA vaccines encoding MERS-CoV antigens (Fig. 7B) were procured from GenScript. The MERS-1227 ferritin mRNA construct encoded the amino acid sequence described above for the ferritin nanoparticle protein. The MERS-1227 cell-anchored mRNA encoded for a MERS-CoV spike containing the England 1 spike residues 1-1332 with a deletion of residues 1228-1295 to reflect the ectodomain deletion present in the MERS-1227 nanoparticle. This construct also included a GGS linker between residue 1227 and the transmembrane domain. The MERS-CoV FL cell-anchored mRNA contained the England 1 residues 1-1332. Both of the cell-anchored mRNA constructs included the 3 proline mutations described above, a mutated furin cleavage site, and a cytoplasmic tail truncation to remove residues 1332-1353, which encode for a cytoplasmic retention motif[97,98]. Codons were optimized for mouse expression. RNAs were synthesized using N1-methyl-pseudouridine, capped with a 5′ Cap1, and stored in 1 mM sodium citrate pH 6.5 buffer until formulation. RNAs were formulated in SM102-containing LNPs at a concentration of 0.1-0.2 mg/mL and stored in PBS pH 7.4, 10% sucrose buffer at -80 °C until use.

### mRNA/LNP cellular potency assays

HeLa cells (ATCC) were plated in a 24-well plate at a density of $2.5 \times 10^5$ cells per well in 450 μL media (RPMI + 10% FBS + 55 μM β-mercaptoethanol) and allowed to adhere for 4 h. LNPs were then diluted to 10X in PBS and added directly to cells in a volume of 50 μL. Cells were incubated with LNPs for 20 h at which point media was removed and cells were treated with StemPro Accutase (Gibco) for 10 min. Detached cells were transferred in technical quadruplicate to a 96-well V-bottom plate and spun at 500 x *g* for 5 min. Media was aspirated and cells were washed once with PBS and then treated with a 1:1000 dilution of Live-or-Dye 665/685 (Biotium) in PBS for 15 min at room temperature. After live/dead staining, two wells per condition were treated with flow buffer (PBS + 1% FBS + 1 mM EDTA) for the "cell-surface expression" condition, while two wells per condition were fixed and permeabilized using Cytofix/Cytoperm (BD) according to the manufacturer's protocol for the "total cell expression" condition. Cells were washed with their respective buffers (flow buffer or 1X Perm/Wash (BD)), then stained with 2.5 μg/mL primary antibodies (JC57-14-hFc and G2-mFc) for 45 min at 4 °C followed by 1 μg/mL of PE-conjugated anti-human secondary antibody (Southern Biotech Cat. No. 2014-09) and Alexa Fluor 405 (AF405)-conjugated anti-mouse secondary antibody (Invitrogen Cat. No. A31553) for 45 min at 4 °C. Cells were then washed in their respective buffers, resuspended in flow buffer, and analyzed on a Cytek Aurora flow cytometer run with Cytek SpectroFlo software. 10,000 events per well were collected and the percentage of positively stained cells in each well was determined relative to a mock-treated condition. Data were analyzed using FlowJo v11 and plotted in GraphPad Prism.

### hDPP4 BALB/c mouse challenge study

DPP4 founder transgenic mice were obtained from Dr. Neeltje van Doremalen at Rocky Mountain Labs (NIH) and used to establish a colony at Colorado State University. Offspring were genotyped as described[76]. Briefly, tail tips from mice were digested and DNA extracted using a commercial kit (Monarch Spin gDNA Extraction kit, NEB) and subjected to conventional PCR using primers that annealed with both the endogenous and transgenic DPP4 alleles (5′-AGCACTTGCTCTCCCAAAGTC) and only the transgenic allele (5′-TCTTCTGTAATCAGCTGCCTTTTA). PCR reactions were electrophoresed on a 2% agarose gel and detection of a 790 bp product indicated the mouse was transgenic (homozygous or heterozygous). DPP4 heterozygotes and homozygotes were used in the challenge study and are indicated in Table S3.

Antigens and adjuvants were formulated to deliver the indicated doses in 100 μL volume. Mice were injected intramuscularly with a 50 μL dose of antigen in each hind leg, to deliver 100 μL of vaccine formulation per immunization. Blood was collected at study days 21 and 42, and sera were stored frozen until analysis. For challenge, mice were lightly anesthetized with ketamine-xylazine and challenged by intranasal instillation of 10,000 PFU of the EMC/2012 strain of MERS-CoV. Mice were weighed and scored for clinical signs of disease daily beginning immediately prior to challenge and extending to euthanasia. Half of the mice were euthanized 3 days post-infection and lungs collected for virus titering. The remainder of the mice were maintained up to day 10 post-infection to assess changes in body weight and survival.

### Ethics statement

All animal studies were conducted in accordance with protocols approved by an Institutional Animal Care and Use Committee (IACUC). WT BALB/c mouse studies were conducted in accordance with the IACUC of Fortis Life Sciences. NHP studies were conducted at Alpha Genesis, Inc. and approved by the Committee on the Care and Use of Laboratory Animal Resources (IACUC Approval: #23-7). Alpaca and human DPP4 mouse challenge studies were approved by the CSU IACUC protocol 4926.

### Reporting summary

Further information on research design is available in the Nature Portfolio Reporting Summary linked to this article.

## Data availability

Data supporting the findings of this study are available from the manuscript and the supplemental information. The mass spectrometry proteomics data have been deposited to the ProteomeXchange Consortium via the PRIDE[99] partner repository with the dataset identifier PXD059802. All source data are provided with this paper in the Source Data file. PDB structures used to generate Fig. 1B are available via accession codes 7X27 and 6XCM. Source data are provided with this paper.

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

## Acknowledgements

We acknowledge receipt of the plasmids used for lentivirus production from BEI Resources (NIH, NIAID), deposited by Dr. Jesse Bloom. We thank Dr. Neeltje van Doremalen for providing the hDPP4 transgenic founder mice. We would like to thank Peter S. Kim for input and advice on project design and influential comments and feedback on this manuscript. We would like to thank Leslie Goo and Jeremy Huynh for helpful comments on this manuscript. We would like to thank Max Li for institutional support throughout the duration of this project. We would like to acknowledge the NIH BioArt and Openclipart resources for the images used in the immunization schemas in this manuscript. Funding for these studies was provided by Vaccine Company, Inc. Additional funding support was provided from the National Institute of Allergy and Infectious Diseases (P01AI167966, DP1AI158186 and 75N93022C00036 to DV) and an Investigators in the Pathogenesis of Infectious Disease Awards from the Burroughs Wellcome Fund (DV). DV is an Investigator of the Howard Hughes Medical Institute and the Hans Neurath Endowed Chair in Biochemistry at the University of Washington.

## Author contributions

A.E.P. and B.A.P. co-led the project team that conducted the preclinical research and development of the MERS-CoV FNPs. P.A.W. designed MERS-CoV FNP constructs. A.E.P. conducted fed-batch CHO productions. A.E.P. and J.O. purified MERS-CoV FNPs. A.E.P., S.P., and B.A.P. designed and conducted stability studies on MERS-CoV FNPs. A.E.P. and B.J.F. expressed and purified antigens for Luminex studies, and J.L.C. designed and performed Luminex binding assays. B.A.P. and J.O. performed LC-MS/MS experiments, and B.A.P. analyzed the peptide mapping data. D.M.B. performed TEM imaging experiments and analysis. H.C. produced MERS-CoV spike-pseudotyped lentiviruses and designed and conducted lentivirus neutralization assays. A.E.P., V.A., J.E.L., M.S.K., P.A.W., and B.A.P. designed mouse and NHP immunization studies. K.R.S., A.A., A.N.A., and D.V. analyzed immunized NHP sera compared to MERS patient samples in the VSV pseudotype neutralization assay. A.E.P. and B.A.P. formulated antigens for mouse, NHP, and alpaca immunizations. D.J.S. conducted ELISpot analyses. V.A. designed and coordinated mRNA procurement for platform evaluation studies, and A.M.W. performed potency characterization assays on mRNA/LNP samples. C.S.D. oversaw the design of immunoassays used for FNP characterization. A.W., A.B., A.H., and R.B. conducted live-virus neutralization assays and an alpaca challenge study. A.E.P. drafted the manuscript with input from B.A.P., M.S.K., and J.E.L. All authors reviewed and edited the manuscript prior to submission.

## Competing interests

A.E.P., S.P., J.O., J.E.L., P.A.W., and B.A.P. are listed as inventors on US patent application 63/587,130, entitled "Engineered Middle East Respiratory Syndrome Proteins and Related Methods" which discloses subject matter described in this paper. A.E.P., H.C., J.L.C., J.O., B.J.F., D.J.S., A.M.W., J.E.L., P.A.W., and B.A.P. are employees of and may hold shares in Vaccine Company, Inc. S.P., V.A., C.S.D., and M.S.K. are former employees of Vaccine Company, Inc. The remaining authors declare no competing interests.
