## [Peer Review file · Nature Communications]

A stabilized MERS-CoV spike ferritin nanoparticle vaccine elicits robust and protective neutralizing antibody responses

Corresponding Author: Dr Abigail Powell

Version 0:

Reviewer comments:

Reviewer #1

(Remarks to the Author)

This manuscript describes two vaccine candidates based on the MERS-CoV spike protein displayed on a self-assembling ferritin nanoparticle. Overall, design elements such as S protein truncation, proline mutations, furin site mutations, flexible linkers, ferritin nanoparticles, and the CHO stable expression system are relatively common. The key contribution of the article lies in its use of multiple animal models, providing substantial preclinical antibody response data.

Major Comments:

1. BLI analysis: The authors confirmed through 2D class results that MERS-1236 is more flexible on the nanoparticle surface. While the BLI results for RBD- and S2-specific antibodies seem to align with this interpretation, how should the results for the NTD-specific monoclonal antibody G2 be explained? Additionally, since the article emphasizes the comparison between MERS-1236 and MERS-1227, could Figure 3 be presented in a comparative format? Otherwise, results under specific conditions, such as stability at 37°C, may be difficult to use as reference indicators for future studies.
2. Vaccination group settings: Why is there no MERS trimer 0.4 µg group in Figure 4? Why is there no placebo group in Figure 5? In Figure 6, why is the MERS-1227 ferritin protein dose set at 0.4 µg, while the mRNA group includes 0.1 and 1 µg doses? Additionally, why is there no MERS-1236 group in Figure 7, as in Figure 8? Does the enhanced flexibility of MERS-1236 on the nanoparticle surface improve epitope presentation, thereby inducing a broader range of neutralizing antibodies and enhancing protective efficacy?
3. Adjuvant selection: Not until Figure 6 do the authors begin comparing how different adjuvant choices influence the immune response characteristics mediated by MERS-1227. This raises the question: What was the rationale behind the adjuvant selection in Figures 4 and 5? Additionally, do the adjuvant evaluation results in Figure 6 provide meaningful insights for current research? Is Alhydrogel truly the optimal choice? Moreover, the adjuvant dosing strategy seems to lack clear logic. Relying solely on the IgG2a/IgG1 ratio to infer cellular immune response characteristics seems insufficient, especially given that the study does not systematically evaluate cellular immunity across its various models.
4. mRNA vaccine: The evaluation of the mRNA vaccine is insufficiently characterized. While the authors constructed three mRNA antigens, the manuscript lacks essential data on the expression of mRNA-LNPs in vivo or in vitro. Additionally, key parameters of the mRNA vaccine should be provided, including particle size, zeta potential, and mRNA-LNP encapsulation efficiency. Why was MERS-1227 chosen? Additionally, why is there no further challenge protection data?

Minor Comments:

1. Please confirm the annotation in Figure 2A. Should the diagram notes "Non-Reduced" and "Reduced" be exchanged? Is it reasonable to run these two types of samples in the same gel?
2. In Figure 2B, please specify the column used in the SEC-MALS method and provide more details on the methods.
3. In Figure 2C, please provide the percentage distribution (pd%) in the DLS test results.
4. In Table S3, it is unusual that the results of PRNT90, PRNT80, and PRNT50 are the same (MERS-1236–Day 42, 0.4 µg dose, MERS-1236–Day 42, 10 µg dose).
5. Add necessary statistical analysis for the related figures.
6. Please verify the errors in the references, such as Ref 66 and Ref 71, to ensure their accuracy and consistency.

Reviewer #2

(Remarks to the Author)

In this manuscript, Abigail et al designed two vaccine candidates, MERS-1227 and MERS-1236, by truncating the

ectodomain of the MERS-CoV spike protein to amino acids 1227 and 1236, respectively. The truncated spike ectodomains were displayed on self-assembled ferritin nanoparticles to improve stability and immunogenicity. In addition, a cytoplasmic tail truncation, mutations of three proline sites and a furin cleavage site were also introduced to stabilize the conformation of the spike protein. After construction and purification, the two vaccine candidates were evaluated with comprehensive characterizations, including size, homogeneity, glycosylation modifications. Even though MERS-1227 displays a more stable functionalized nanoparticle, both of MERS-1227 and MERS-1236 are able to maintain antigenicity under pH 7.5 or 37 °C conditions for 14 days. Both candidates elicit robust neutralizing antibody responses, especially after a second boost dose, in mice and non-human primate models, and exhibit obviously protective functions against virus infection in hDPP4 transgenic mice and alpacas models. Overall, this work is the first model of MERS nanoparticle vaccine that achieves complete protection in large animals and has significant clinical development value. The evaluation is systematic and comprehensive, with solid data. The publication is recommended after minor revision: glycosylation is critical for protein conformation and stabilization, and also affect the recognition between virus and receptors. Thus, glycosylation modifications on protein are important in vaccine production. It is recommended to add more detailed information of glycosylation modifications (e.g., glycan compositions and their abundance distribution at each site) on MERS-1227 and MERS-1236. And a minor mistake in line 429 should be addressed: "Figure 7F" should be corrected to "Figure 7E".

Reviewer #3

(Remarks to the Author)

Summary of key results:

The manuscript describes the design and production of recombinant MERS vaccine antigen MERS-1227 (and the alternative MERS-1236), a stabilized (truncated) spike ectodomain displayed on a self-assembling ferritin nanoparticle. The antigen is evaluated as a vaccine candidate in different animal models. The authors compare its immunogenicity in mice using different adjuvants. They also test the formulation with alum in non-human primates (as a human model) and assess efficacy in both mice and alpacas (as a dromedary model).

Validity and significance:

Overall, the manuscript presents a substantial amount of high-quality data on the vaccine antigen and its various formulations. In my opinion, these findings hold significant value for the scientific community. Moreover, the open publication of this data would be particularly valuable if the antigen proceeds to clinical trials, as it would provide important context and transparency for future development.

Clarity and context:

The manuscript is well-written, including the methods section, which contains all the necessary details. Results are adequately discussed.

Comments and Suggestions:

- The study seems to focus exclusively on the humoral immune response, specifically the induction of neutralizing antibodies. As seen with other coronaviruses, long-term protection, particularly against severe disease, may rely more on cellular immune responses. In my opinion, it is relevant to also compare the cellular response (e.g., cytokine release after antigen-specific stimulation of T cells). This could easily have been done for the mouse study in the context of adjuvant selection (e.g., alum vs. others) and platform comparisons (e.g., recombinant protein vs. mRNA vaccines).
- The authors appear to favor the alum-adjuvanted MERS-1227 based on the use of the adjuvant in the vaccine challenge test. However, Quil-A + MPLA had a 22-fold better NT50 after 2 doses. Comparing the most immunogenic formulation to the alum formulation in the challenge models would have been interesting.
- I would further caution that the alum-based Th2-skewed immune response may be less effective in preventing severe disease compared to vaccines that elicit a Th1-biased or more balanced Th1/Th2 response. Although the authors mention, "Taken together, the results of this adjuvant screen demonstrate several feasible mechanisms to improve both humoral and cell-mediated immune responses to MERS-1227", the important role of cell-mediated immune responses is not well discussed in the manuscript.

Minor comments:

- Fig 6 panel B, replace +/- with the actual dose of alum in ug (0 and 150)
- The numbering of the references is off by 1; I believe reference 39 is missing in the main text.

Reviewer #4

(Remarks to the Author)

In this study, Powell and colleagues described the design, characterization, and preclinical evaluation of two self-assembling ferritin nanoparticle-based MERS-CoV Spike vaccine candidates, MERS-1227 and MERS-1236. While ferritin nanoparticle vaccines for coronaviruses have been previously explored, the authors enhance the structural uniformity and thermodynamic stability of their candidates through optimized manufacturing processes. However, the antigen design lacks sufficient novelty, and some critical data of protection effect of the preclinical evaluation of vaccine candidates on several animal models are missing. Moreover, several key issues need to be addressed and clarified.

1. MERS-1227 exhibits higher structural stability and immunogenicity than MERS-1236. Could the authors provide structural insights (e.g., cryo-EM analysis or molecular dynamics simulations) to explain how the amino acid truncation contributes to these differences?
2. Cellular immune responses also play a critical role in protection against MERS-CoV infection, it is suggested to evaluate cellular immune responses in vaccinated animals.
3. Additional well-characterized MERS-CoV monoclonal antibodies are recommended to enroll for nanoparticles binding to further confirm proper antigenic site exposure.
4. How do these ferritin nanoparticle vaccines compare to other MERS-CoV vaccine candidates (e.g., adenoviral vectors, mRNA-LNPs, or subunit vaccines) in terms of neutralizing antibody titers, durability, or cross-reactivity?
5. The absence of histopathological analysis (e.g., lung tissue scoring for inflammation, viral antigen distribution) in the challenge studies is notable, which will strengthen the correlation between neutralizing antibody levels and actual protection.

Version 1:

Reviewer comments:

Reviewer #1

(Remarks to the Author)

The authors have thoroughly addressed the comments from the first round of peer review and provided satisfactory additional data and clarifications on key aspects such as adjuvant selection, glycosylation profiling, and cellular immune responses. I recommend acceptance pending minor revisions.

Minor comments:

1. The statement of Figure S4 is clearer than Figure 3, and it is recommended to add Figure S4 to the main image.
2. Line 1468, figure 2E, TEM micrographs are missing clear scale.
3. Line 1493, Figure S3, Figure Note is missing.

Reviewer #3

(Remarks to the Author)

The authors did a substantial effort to address most of the reviewer comments. The data of the cellular data of the restimulated PBMC samples from the NHP study is a valuable addition.

I recommend publication

One very minor request can be further addressed

A reviewer asked to provide the percentage distribution (pd%) in the DLS test results, probably referring to the polydispersity (PDI). I agree this could be of value to indicate monodispersity and can be added in the figure legend with the size estimate. That number is certainly retrievable from a DLS dataset of the Prometheus Panta.

Reviewer #4

(Remarks to the Author)

In this revised manuscript, critical data have been updated with an optimized visual presentation by Powell and colleagues. However, the revised version has not addressed the key issues raised by the reviewers.

1. Critical histopathological experiments remain unprovided.
2. The supplementary T cell response experiments indicate that the vaccine induces a Th2-dominated T cell response, raising the question of whether this could lead to more severe inflammation.

Version 2:

Reviewer comments:

Reviewer #1

(Remarks to the Author)

No more concerns

REVIEWER COMMENTS

Reviewer #1 (Remarks to the Author):

This manuscript describes two vaccine candidates based on the MERS-CoV spike protein displayed on a self-assembling ferritin nanoparticle. Overall, design elements such as S protein truncation, proline mutations, furin site mutations, flexible linkers, ferritin nanoparticles, and the CHO stable expression system are relatively common. The key contribution of the article lies in its use of multiple animal models, providing substantial preclinical antibody response data.

Major Comments:

1. BLI analysis

The authors confirmed through 2D class results that MERS-1236 is more flexible on the nanoparticle surface. While the BLI results for RBD- and S2-specific antibodies seem to align with this interpretation, how should the results for the NTD-specific monoclonal antibody G2 be explained? Additionally, since the article emphasizes the comparison between MERS-1236 and MERS-1227, could Figure 3 be presented in a comparative format? Otherwise, results under specific conditions, such as stability at 37°C, may be difficult to use as reference indicators for future studies.

We thank the reviewer for this comment regarding the impact of protein flexibility on the BLI binding results. To better understand this relationship, we expanded our mAb panel from 4 to 12. In this expanded panel, we assess 11 antibodies targeting the S1 domain, 8 which target the RBD, and 3 which target non-RBD epitopes. All 8 RBD antibodies show enhanced binding to MERS-1236, whereas the 3 non-RBD S1 antibodies exhibit comparable MERS-1227 and MERS-1236 binding. This suggests to us that the major impact of this increased flexibility is enhanced dynamics of the RBD sampling the known “up” and “down” conformations. The enhanced binding to 3A3, an S2-targeting antibody, could be suggestive of overall increased trimer breathing in the MERS-1236 construct, however without additional available S2 antibodies we are unable to further profile this.

We agree with the reviewer that an alternative presentation of the stability data would be helpful and have added a supplemental figure (Figure S4) with the relevant subset of the data presented in a comparative manner between MERS-1227 and MERS-1236. We have also included all raw data in the source data file so that additional comparisons of interest could be plotted.

Reviews received: 15 April 2025
Responses submitted: 15 July 2025

2. Vaccination group settings

We thank the reviewer for this comment regarding the dosing and study design for our preclinical immunization experiments. We address these comments by individual point below and have made clarifying changes to the text, figures, and figure order as noted.

Why is there no MERS trimer 0.4 µg group in Figure 4?

We agree that it added confusion to include the 0.4 µg dose group in Figure 4 for the MERS-1227 and MERS-1236 groups but not the MERS-CoV trimer. At the time of this study, we did not have the resources to include a 0.4 µg dose group for the trimer and therefore we do not have these data. We have removed the 0.4 µg dose group for MERS-1227 and MERS-1236 from Figure 4B-C for increased clarity. We have also amended the text accordingly.

Why is there no placebo group in Figure 5?

The inclusion of a placebo group in this experiment would have been prohibitively expensive. However, as a control, we evaluated study day 0 (pre-immune) serum in pseudovirus neutralization and Luminex binding assays. This serum is from the same NHPs as was taken at the later time points, which demonstrates the effects of vaccination on the corresponding antibody responses. The pre-immune serum shows no quantifiable binding or neutralization signals, indicating that the responses seen at later time points are the result of the immune response to immunization.

In Figure 6, why is the MERS-1227 ferritin protein dose set at 0.4 µg, while the mRNA group includes 0.1 and 1 µg doses?

As the work presented here was primarily focused on characterization of the MERS-CoV protein nanoparticles, we conducted a larger set of preclinical analyses using this antigen as compared to the mRNA versions of these antigens. We chose 0.4 µg protein dose based on our dose de-escalation study shown in Figure S6, which showed that below 0.4 µg, there was a dose-dependent reduction in antibody responses to the protein. We were unable to perform this thorough dosing evaluation on all antigens. As the mRNA and protein nanoparticles are distinct platforms, we did not expect that comparing corresponding dose levels would necessarily be meaningful. Instead, we evaluated the mRNAs at two commonly used dose levels utilized in preclinical mouse mRNA studies in the literature (0.1 and 1 µg)¹.

Additionally, why is there no MERS-1236 group in Figure 7, as in Figure 8?

This comment made us recognize that the figure ordering could have improved clarity. While both MERS-1227 and MERS-1236 have promising attributes and are largely equivalent immunogenically, we ultimately down-selected to MERS-1227 as the lead development candidate based on concerns surrounding the flexibility and lower melting temperature of the MERS-1236 construct. As the adjuvant, platform, and DPP4 mouse

Reviews received: 15 April 2025
Responses submitted: 15 July 2025

challenge study required extensive resources, we conducted them after this down-selection. To make the logic behind this selection clearer, we have added clarifying text and moved the alpaca challenge study (the last major study containing both MERS-1227 and MERS-1236) to follow the NHP study (Figure 5).

Does the enhanced flexibility of MERS-1236 on the nanoparticle surface improve epitope presentation, thereby inducing a broader range of neutralizing antibodies and enhancing protective efficacy?

This is an interesting hypothesis; however, we do not think the neutralization of our MERS-CoV viral panel in Figure 5E are conclusive enough to state this in the results. We have added text in the discussion raising this as a possibility.

3. Adjuvant selection

Not until Figure 6 do the authors begin comparing how different adjuvant choices influence the immune response characteristics mediated by MERS-1227. This raises the question: What was the rationale behind the adjuvant selection in Figures 4 and 5?

Additionally, do the adjuvant evaluation results in Figure 6 provide meaningful insights for current research? Is Alhydrogel truly the optimal choice? Moreover, the adjuvant dosing strategy seems to lack clear logic. Relying solely on the IgG2a/IgG1 ratio to infer cellular immune response characteristics seems insufficient, especially given that the study does not systematically evaluate cellular immunity across its various models.

We thank the reviewer for this feedback pertaining to adjuvant usage. The rationale for using alum in the initial immunogenicity studies presented in Figures 4 and 5 stemmed from its widespread clinical use and the ability to readily procure this adjuvant for a clinical development program (lines 388-390). We reasoned that showing strong immunogenicity with a mild adjuvant like alum would provide proof of concept to promote further development of these vaccine candidates. The adjuvant screening results are intended to demonstrate that our vaccine is compatible with and can potentially be enhanced by alternative adjuvants relative to alum. We hope these results may be further tested in the future, perhaps with a bedside mixing strategy in humans, if the proprietary versions of these adjuvants can be accessed for human use.

We agree with the reviewer that the IgG2a/IgG1 ratio does not fully address cellular immune responses. We have now assessed vaccinated samples from our NHP study to evaluate IFN γ and IL-4 responses, presented in Figure 5F and S7C. Since we had not collected mouse spleens, we are unable to retrospectively assess the cellular immune responses in those animals. We have tempered the language around this conclusion (lines 427-429), but we do hope the reviewer agrees that we fulfilled the primary goal of optimizing neutralizing antibody responses.

Reviews received: 15 April 2025
Responses submitted: 15 July 2025

4. mRNA vaccine

The evaluation of the mRNA vaccine is insufficiently characterized. While the authors constructed three mRNA antigens, the manuscript lacks essential data on the expression of mRNA-LNPs in vivo or in vitro. Additionally, key parameters of the mRNA vaccine should be provided, including particle size, zeta potential, and mRNA-LNP encapsulation efficiency. Why was MERS-1227 chosen? Additionally, why is there no further challenge protection data?

We thank the reviewer for this comment. We have added both biophysical (Figure S9A) and in vitro cellular potency (Figure S9B-C) data to describe these results and added associated discussion in the main text (lines 442-450). The biophysical characterization includes % mRNA encapsulation, size, polydispersity, and zeta potential and the cellular potency includes both cell-surface and total cell expression.

We added text in the manuscript (lines 380-385) to indicate that following our alpaca challenge results, we down-selected the candidates to focus on MERS-1227 based on the comparable immunogenicity and the more favorable stability profile of MERS-1227. This was largely driven by potential manufacturing concerns due to the flexibility and lower melting temperature of MERS-1236. For this reason, when we progressed to the adjuvant screen and the mRNA platform evaluation, we solely evaluated MERS-1227.

Minor Comments:

1. Please confirm the annotation in Figure 2A. Should the diagram notes "Non-Reduced" and "Reduced" be exchanged? Is it reasonable to run these two types of samples in the same gel?

We thank the reviewer for noting this and confirm that the labels in Figure 2A are correct. Given the disulfide bonds in the MERS-CoV spike, our interpretation here is that releasing those bonds in the protein lead to an extended conformation that runs more slowly via SDS-PAGE (noted in lines 176-177 in the text). Since the reducing agent (beta-mercaptoethanol) is added directly to the sample loading buffer (not the gel or the running buffer), we think it was appropriate to run these on the same gel.

2. In Figure 2B, please specify the column used in the SEC-MALS method and provide more details on the methods.

We have added both the column type and the light scattering detector name to figure legend 2B.

3. In Figure 2C, please provide the percentage distribution (pd%) in the DLS test results.

Reviews received: 15 April 2025
Responses submitted: 15 July 2025

The instrument used to collect these results does not report on the percent distribution of the results and thus unfortunately we are unable to add this parameter into our reporting.

4. In Table S3, it is unusual that the results of PRNT90, PRNT80, and PRNT50 are the same (MERS-1236–Day 42, 0.4 µg dose, MERS-1236–Day 42, 10 µg dose).

This neutralization assay was conducted by limiting dilution, and expressed as the dilution point at which 90% (PRNT90), 80% (PRNT80), or 50% (PRNT50) of viral plaques were inhibited. Serum was tested from a 1:10 to a 1:320 dilution. We have removed the discussion of the 0.4 µg dose of MERS-1236 to be consistent with revisions made to groups in Figure 4. For the 10 µg dose, the ≥ 320 indicated that at a dilution greater than or equal to 1:320, this serum inhibited 50%, 80%, and 90% of viral plaques and thus was at the upper limit of quantitation of this assay. We have adjusted the symbols from \geq to \geq for clarity. We have also added text to the methods section (lines 959-962) further describing this live virus neutralization method.

5. Add necessary statistical analysis for the related figures.

We have noted the statistical analyses performed in the figure legends (Figures 4B-C) and in the corresponding methods sections (lines 832-833 and 866-867).

6. Please verify the errors in the references, such as Ref 66 and Ref 71, to ensure their accuracy and consistency.

We thank the reviewer for spotting this error and have corrected these references.

Reviewer #2 (Remarks to the Author):

In this manuscript, Abigail et al designed two vaccine candidates, MERS-1227 and MERS-1236, by truncating the ectodomain of the MERS-CoV spike protein to amino acids 1227 and 1236, respectively. The truncated spike ectodomains were displayed on self-assembled ferritin nanoparticles to improve stability and immunogenicity. In addition, a cytoplasmic tail truncation, mutations of three proline sites and a furin cleavage site were also introduced to stabilize the conformation of the spike protein. After construction and purification, the two vaccine candidates were evaluated with comprehensive characterizations, including size, homogeneity, glycosylation modifications. Even though MERS-1227 displays a more stable functionalized nanoparticle, both of MERS-1227 and MERS-1236 are able to maintain antigenicity under pH 7.5 or 37 °C conditions for 14 days. Both candidates elicit robust neutralizing antibody responses, especially after a second boost dose, in mice and non-human primate models, and exhibit obviously protective functions against virus infection in hDPP4 transgenic mice and alpacas models. Overall, this work is the first model of MERS nanoparticle vaccine that achieves complete protection in large animals and has

Reviews received: 15 April 2025
Responses submitted: 15 July 2025

significant clinical development value. The evaluation is systematic and comprehensive, with solid data.

The publication is recommended after minor revision: glycosylation is critical for protein conformation and stabilization, and also affect the recognition between virus and receptors. Thus, glycosylation modifications on protein are important in vaccine production.

It is recommended to add more detailed information of glycosylation modifications (e.g., glycan compositions and their abundance distribution at each site) on MERS-1227 and MERS-1236.

We agree that analysis of the glycosylation of our antigens could affect their antigenicity and that this is an important parameter to monitor and control during vaccine production. In the original version of the manuscript, we performed LC-MS/MS glycopeptide analysis and concluded that both MERS-1227 and MERS-1236 had extensive N-glycosylation and limited but detectable O-linked glycosylation (Figure S2 and Table S1). The glycan compositions were included in Table S1 of the original manuscript, but we did not clearly state this in the main text and thank the reviewer for bringing this to our attention. In the revised manuscript, we have more clearly noted this analysis in the Results, lines 183-190. Additionally, we have performed the additional data analysis requested by the reviewer and now report the relative ion abundances of each glycan at each site, relative to the total ion abundance at that site. This expanded analysis is also mentioned in the Results (lines 184-187).

And a minor mistake in line 429 should be addressed: “Figure 7F” should be corrected to “Figure 7E”.

Thank you for noting this, we have corrected this in the text.

Reviewer #3 (Remarks to the Author):

Summary of key results:

The manuscript describes the design and production of recombinant MERS vaccine antigen MERS-1227 (and the alternative MERS-1236), a stabilized (truncated) spike ectodomain displayed on a self-assembling ferritin nanoparticle. The antigen is evaluated as a vaccine candidate in different animal models. The authors compare its immunogenicity in mice using different adjuvants. They also test the formulation with alum in non-human primates (as a human model) and assess efficacy in both mice and alpacas (as a dromedary model).

Validity and significance:

Overall, the manuscript presents a substantial amount of high-quality data on the vaccine antigen and its various formulations. In my opinion, these findings hold significant value for the scientific community. Moreover, the open publication of this data

Reviews received: 15 April 2025
Responses submitted: 15 July 2025

would be particularly valuable if the antigen proceeds to clinical trials, as it would provide important context and transparency for future development.

Clarity and context:

The manuscript is well-written, including the methods section, which contains all the necessary details. Results are adequately discussed.

Comments and Suggestions:

- The study seems to focus exclusively on the humoral immune response, specifically the induction of neutralizing antibodies. As seen with other coronaviruses, long-term protection, particularly against severe disease, may rely more on cellular immune responses. In my opinion, it is relevant to also compare the cellular response (e.g., cytokine release after antigen-specific stimulation of T cells). This could easily have been done for the mouse study in the context of adjuvant selection (e.g., alum vs. others) and platform comparisons (e.g., recombinant protein vs. mRNA vaccines).

We thank the reviewer for this comment and agree that cellular immune assessments can provide a more comprehensive evaluation of our vaccine. Since we did not collect mouse spleen samples, we were unable to retrospectively evaluate cellular immune responses in these animals. However, we did have PBMC samples from our NHP study. Thus, we have added both IFN γ and IL-4 ELISpot results in Figures 5F and S7C demonstrating that both MERS-1227 and -1236 elicit T cell responses against both MERS-CoV spike and the *H. pylori* ferritin scaffold.

- The authors appear to favor the alum-adjuvanted MERS-1227 based on the use of the adjuvant in the vaccine challenge test. However, Quil-A + MPLA had a 22-fold better NT50 after 2 doses. Comparing the most immunogenic formulation to the alum formulation in the challenge models would have been interesting.

We thank the reviewer for this comment. The basis for our focus on alum is rooted in the availability of alum as a clinical adjuvant as it poses substantially fewer hurdles for use as a clinical development candidate compared to proprietary adjuvants such as those utilized in our adjuvant screen. We agree that future challenge studies with the most immunogenic formulations could be informative, however this was outside the scope of this current body of work.

- I would further caution that the alum-based Th2-skewed immune response may be less effective in preventing severe disease compared to vaccines that elicit a Th1-biased or more balanced Th1/Th2 response. Although the authors mention, "Taken together, the results of this adjuvant screen demonstrate several feasible mechanisms to improve both humoral and cell-mediated immune responses to MERS-1227", the important role of cell-mediated immune responses is not well discussed in the manuscript.

Reviews received: 15 April 2025
Responses submitted: 15 July 2025

We thank the reviewer for noting this. As stated above, we have added T cell analysis of our vaccinated NHP samples into the manuscript. We have also modified the language at the end of the adjuvant screen results section to soften the conclusion on cellular immunity to read: "Taken together, the results of this adjuvant screen demonstrate that modification of the adjuvant formulation of MERS-1227 could be used to enhance the resultant antibody responses and potentially modulate the cellular immune response." (lines 427-429)

Minor comments:

- Fig 6 panel B, replace +/- with the actual dose of alum in ug (0 and 150)

We have made this change to this figure (which is now Figure 7).

- The numbering of the references is off by 1; I believe reference 39 is missing in the main text.

We have corrected these errors with the references.

Reviewer #4 (Remarks to the Author):

In this study, Powell and colleagues described the design, characterization, and preclinical evaluation of two self-assembling ferritin nanoparticle-based MERS-CoV Spike vaccine candidates, MERS-1227 and MERS-1236. While ferritin nanoparticle vaccines for coronaviruses have been previously explored, the authors enhance the structural uniformity and thermodynamic stability of their candidates through optimized manufacturing processes. However, the antigen design lacks sufficient novelty, and some critical data of protection effect of the preclinical evaluation of vaccine candidates on several animal model are missing. Moreover, several key issues need to be addressed and clarified.

1. MERS-1227 exhibits higher structural stability and immunogenicity than MERS-1236. Could the authors provide structural insights (e.g., cryo-EM analysis or molecular dynamics simulations) to explain how the amino acid truncation contributes to these differences?

We thank the reviewer for this comment. We conducted a preliminary follow-up cryo-TEM imaging analysis which revealed similar 2D class averaging results to our negative stain TEM, albeit with substantial smearing of the spike density surrounding the ferritin core, likely due to the high flexibility of these molecules. For this reason, we concluded that further structural image analysis would be unlikely to reveal additional details about these molecules given their size and flexibility. We currently do not have the computation resources to run molecular dynamics simulations on these molecules and feel that gaining access to this analysis would be outside the scope of this study. However, we have provided the amino acid sequences of MERS-1227 and MERS-1236

Reviews received: 15 April 2025
Responses submitted: 15 July 2025

in Table S2 to allow such MD simulations to be conducted independently by groups with these resources, if desired.

2. Cellular immune responses also play a critical role in protection against MERS-CoV infection, it is suggested to evaluate cellular immune responses in vaccinated animals.

We thank the reviewer for this comment. We have now evaluated T cell responses in the NHPs vaccinated with alum-adjuvanted MERS-1227 and MERS-1236. These results have been added in Figures 5F and S7C and demonstrate both IFN γ and IL-4 responses to both candidates following both 1 and 2 doses of vaccine.

3. Additional well-characterized MERS-CoV monoclonal antibodies are recommended to enroll for nanoparticles binding to further confirm proper antigenic site exposure.

We thank the reviewer for this comment and in response have expanded our mAb panel for characterization from 4 antibodies to 12 antibodies, presented in Figures 2F and S3. With this expanded panel, we can conclude that the differences in binding to MERS-1227 and MERS-1236 are specific for the RBD within the S1, given that 3 non-RBD S1-targeting antibodies show equivalent binding between the two antigens.

4. How do these ferritin nanoparticle vaccines compare to other MERS-CoV vaccine candidates (e.g., adenoviral vectors, mRNA-LNPs, or subunit vaccines) in terms of neutralizing antibody titers, durability, or cross-reactivity?

While a number of MERS vaccine candidates have been evaluated preclinically, there is a lack of consistent assay methodology to appropriately compare across different studies. We were unable to find reports of either preclinical or clinical MERS candidates that evaluated either vaccine durability. We found three preclinical vaccine candidates (a DNA vaccine, an Adeno-based vaccine, and a protein nanoparticle vaccine) that evaluated cross-reactive antibody responses and have added this in the discussion (lines 548-550).

As an alternative means of benchmarking our reported pseudovirus neutralization titers, we included a new set of neutralization analyses assessing titers from our immunized NHPs compared with a set of human convalescent patient serum samples. All samples were analyzed concurrently in a VSV pseudotyped neutralization assay (Figure 5D). We were encouraged that our single-dose titers are comparable to convalescent patient titers and after a boost, we observe titers ~100-fold higher.

5. The absence of histopathological analysis (e.g., lung tissue scoring for inflammation, viral antigen distribution) in the challenge studies is notable, which will strengthen the correlation between neutralizing antibody levels and actual protection.

Reviews received: 15 April 2025
Responses submitted: 15 July 2025

We thank the reviewer for this comment. While we were unable to conduct these analyses for our currently reported challenge studies, we agree that future challenge experiments would benefit from this expanded set of data.

References

1. Corbett, K. S. *et al.* SARS-CoV-2 mRNA vaccine design enabled by prototype pathogen preparedness. *Nature* **586**, 567–571 (2020).
2. Watanabe, Y. *et al.* Vulnerabilities in coronavirus glycan shields despite extensive glycosylation. *Nat Commun* **11**, 2688 (2020).

REVIEWER COMMENTS

Reviewer #1 (Remarks to the Author):

The authors have thoroughly addressed the comments from the first round of peer review and provided satisfactory additional data and clarifications on key aspects such as adjuvant selection, glycosylation profiling, and cellular immune responses. I recommend acceptance pending minor revisions.

Minor comments:

1.The statement of Figure S4 is clearer than Figure 3, and it is recommended to add Figure S4 to the main image.

We thank the reviewer for this comment and agree the representation of these results in Figure S4 were more cohesive with the remainder of the data presented in the main text. We have switched Figure S4 with Figure 3. All data are available in the Source File if readers want to make any additional comparative analyses.

2.Line 1468, figure 2E, TEM micrographs are missing clear scale.

We thank the reviewer for noting this, we have added the scale in text below the scale bar.

3.Line 1493, Figure S3, Figure Note is missing.

We thank the reviewer for this comment; however, we are not sure what the reviewer is requesting. In our version, the Figure Legend for Figure S3 reads:

“Figure S3. BLI binding of a panel of MERS-CoV mAbs to MERS-1227 and MERS-1236. mAbs were loaded onto anti-human capture sensor tips and dipped into either MERS-1227 or MERS-1236 wells at 100 µg/mL FNP. Dashed line indicates the end of the association step.”

at line 1493.

Reviewer #3 (Remarks to the Author):

The authors did a substantial effort to address most of the reviewer comments. The data of the cellular data of the restimulated PBMC samples from the NHP study is a valuable addition. I recommend publication

One very minor request can be further addressed

Reviewer comments received: 20 August 2025

Responses submitted: 05 November 2025

A reviewer asked to provide the percentage distribution (pd%) in the DLS test results, probably referring to the polydispersity (PDI). I agree this could be of value to indicate monodispersity and can be added in the figure legend with the size estimate. That number is certainly retrievable from a DLS dataset of the Prometheus Panta.

We thank the reviewer for this comment. We have added a table in Figure 2C including both the diameter and PDI measurements from this DLS analysis.

Reviewer #4 (Remarks to the Author):

In this revised manuscript, critical data have been updated with an optimized visual presentation by Powell and colleagues. However, the revised version has not addressed the key issues raised by the reviewers.

1. Critical histopathological experiments remain unprovided.

While we are unable to conduct additional challenge experiments at this time, we agree with the reviewer that histopathological data, including lung tissue scoring and viral antigen distribution, are a key metric in these studies. Upon discussion with our collaborating authors who conducted the challenge work, we learned that lung pathology was evaluated for all mice included in the hDPP4 mouse challenge study. We have added these lung pathology scores into table S5 and described these results in the text. While minimal lung pathology was observed in both the vaccinated and placebo-treated mice, lung inflammation and histopathological changes were also not noted in the publication which established the hDPP4 mouse model of use¹. Though we did not specifically look at viral antigen distribution, our collaborating authors did assess virus levels in the lung using plaque assay. These results are presented in Figure 8D and demonstrate that the vaccinated mice have lower levels of replicating virus isolated from lung tissue following the viral challenge. This correlates well with production of neutralizing antibodies in these animals, as presented in Figure 8B-C. We have added text highlighting these results and have also added the numerical viral titers in Table S5.

For the alpaca challenge study, this work was conducted in a specialized large-animal BSL3 facility and due to the complexities for this working environment, our collaborating authors were unable to conduct extensive necropsy following the virus challenge.

2. The supplementary T cell response experiments indicate that the vaccine induces a Th2-dominated T cell response, raising the question of whether this could lead to more severe inflammation.

We thank the reviewer for noting this important point. While we are unable to gather experimental data to confirm or refute whether the Th2-dominated response in our animal studies could lead to severe inflammation, we have noted in the Results when describing the

Reviewer comments received: 20 August 2025

Responses submitted: 05 November 2025

NHP PBMC evaluation (lines 357-362) and the hDPP4 mouse challenge study (lines 491-498) the potential for Th2-induced inflammation and concerns surrounding this phenomenon. While our hDPP4 mouse lung necropsy results did not suggest vaccine-induced lung inflammation, we have caveated these results in the text indicating that future studies with more in-depth histopathology evaluation would be required to fully rule this out. We also highlighted the possibility of immunopathology following a Th2-biased vaccine-induced immune response in the Discussion (lines 570-575) as an area for future evaluation, likely in the context of modified adjuvant formulations.

References

1. van Doremalen, N. *et al.* Transmission dynamics of MERS-CoV in a transgenic human DPP4 mouse model. *npj Viruses* **2**, 1–8 (2024).